# The Triassic turtle of Thailand – revision of *'Proganochelys' ruchae*

**Tomasz Szczygielski**[1]*, **Dawid Dróżdż**[2], **Phornphen Chanthasit**[3], **Sita Manitkoon**[4,5], **Pitaksit Ditbanjong**[6]

**1** Institute of Paleobiology, Polish Academy of Sciences, Warsaw, Poland, **2** Independent Researcher, Cieszyn, Poland, **3** Department of Mineral Resources, Sirindhorn Museum, Kalasin, Thailand, **4** Palaeontological Research and Education Centre, Mahasarakham University, Maha Sarakham, Thailand, **5** Vertebrate Palaeontology and Evolution Research Unit, Excellence Center in Basin Studies and Applied Paleontology, Mahasarakham University, Kham Riang, Thailand, **6** Department of Geotechnology, Faculty of Technology, Khon Kaen University, Khon Kaen, Thailand

* t.szczygielski@twarda.pan.pl

## Abstract

True turtles (*Testudinata*) appeared in the Norian (Late Triassic) and quickly attained a worldwide distribution and relatively high diversity. Their remains are currently known from that time from Asia, Europe, North America, and South America, and represent at least three separate clades. Whereas the generic and suprageneric attribution of comparatively well-preserved and studied European and South American taxa, such as *Proganochelys quenstedtii*, *Proterochersis robusta* and *Proterochersis porebensis*, *Palaeochersis talampayensis*, and *Waluchelys cavitesta* sparks no controversy, the more fragmentary and less common species have been variably considered representatives of separate genera or referred to already existing ones – most commonly, *Proganochelys*. This uncertainty is unfortunate, as it severely muddles the picture of the evolution, diversity, and geographical distribution of the earliest turtles. One such problematic species, coming from the Norian Huai Hin Lat Formation of Thailand, was described in 1980s as *Proganochelys ruchae*. However, this generic attribution was subsequently put into question and the recent increase of available Triassic turtle material allows to remove the species *ruchae* from *Proganochelys* to a new genus, *Thaichelys* gen. nov., and place it in the clade of *Proterochersidae*, together with *Proterochersis* spp., *Keuperotesta limendorsa*, and *Chinlechelys tenertesta*. As a result, the genus *Proganochelys* is considered here to be solely a central Pangean (modern-day European) taxon. Moreover, *Thaichelys ruchae* exhibits in some respects a transitional morphology between other Triassic taxa and *Proterochersis* spp., and may constitute a sister taxon to the grouping of *Proterochersis* spp. and *Keuperotesta limendorsa* from Europe. This, in turn, considering the lack of a Carnian record of pantestudinates outside of Asia, suggests that *Thaichelys ruchae* could represent an early radiation of the *Proterochersidae* which branched off before the *Testudinata* dispersed into the western Pangea.

## Introduction

The first work presenting turtle remains from the Triassic of Thailand was published by Broin *et al.* [1], who described four shell fragments from the Late Triassic (Norian) Huai Hin Lat

**Data availability statement:** All the generated 3D models are accessible for download from MorphoSource: https://www.morphosource. org/projects/000683358

**Funding:** The study was supported by the National Science Centre, Poland (Narodowe Centrum Nauki) grant no. 2020/39/B/ NZ8/01074 awarded to T. Sz. The funders did not play any role in the study design, data collection and analysis, decision to publish, or preparation of the manuscript.

**Competing interests:** The authors have declared no competing interests exist.

**Institutional abbreviations** CSMM, Carl-Schweizer-Museum Murrhardt, Murrhardt, Germany; IVPP, Institute of Vertebrate Paleontology and Paleoanthropology, Chinese Academy of Sciences, Beijing, China; MB, Museum für Naturkunde, Berlin, Germany; MNHN, Muséum National d'Histoire Naturelle, Paris, France; NHMD, Natural History Museum of Denmark, Copenhagen, Denmark; NMMNH, New Mexico Museum of Natural History and Science, Albuquerque, USA; PRC, Palaeontological Research and Education Centre, Mahasarakham University, Mahasarakham, Thailand (former CY); PULR, Universidad Nacional de La Rioja, La Rioja, Argentina; PVSJ, Paleontología de Vertebrados, Museo de Ciencias Naturales de San Juan, San Juan, Argentina; SM, Sirindhorn Museum, Department of Mineral Resources, Kalasin, Thailand (former TF, Department of Mineral Resources, Bangkok, Thailand); SMF, Sauriermuseum Frick, Frick, Switzerland; SMNS, Staatliches Museum für Naturkunde Stuttgart, Stuttgart, Germany; ZPAL, Institute of Paleobiology, Polish Academy of Sciences, Warsaw, Poland.

Formation, but also announced discovery of more, better preserved bones, which at the time were still being prepared. The latter were documented two years later and, together with the previous material, served to establish the new species, *Proganochelys ruchae* (Broin, 1984) [2]. The referral to the genus *Proganochelys* Baur, 1887 [3] was mostly based on the presence of profound gular and extragular projections and dorsal epiplastral processes, and the species was distinguished from *Proganochelys quenstedtii* Baur, 1887 [3] based on the lateral rather than anterior direction of the distal points of the extragular processes and less distinct shell ornamentation. These initial descriptions of the Triassic turtle material from Thailand, however, were published not only before the pivotal monograph of Eugene S. Gaffney [4] on *Proganochelys quenstedtii*, but also before the subsequent discoveries of Triassic turtles from South America [5–7] and North America [8–10], Chinese and German non-turtle pantestudinates [11–15], as well as the reassessment of European proterochersid turtle finds [16–23]. Broin [2] was aware that data on Triassic turtle anatomy and evolution were sparse at the time and clearly stated that the inclusion of the new species in the genus *Proganochelys* is provisional, and indeed, the ensuing gradual increase of the knowledge on Triassic turtles led to doubts about its identity. Gaffney [24] and Lapparent de Broin and Murelaga [25] referred to the species as '*Proganochelys*' *ruchae*, Lapparent de Broin [26,27] and Szczygielski and Piechowski [28] listed it as aff. *Proganochelys ruchae*, and Lapparent de Broin *et al.* [29] as a form affine to *Proganochelys*. Despite those doubts, the material was never revised in detail. Numerous authors [5–7,10,19,20,30–33] repeated the original attribution and Joyce [34] accepted the species as a member of *Proganochelys*, again citing the presence of well-developed gular projections as diagnostic, and the more lateral alignment of the extragular projections as differentiating the species *ruchae* from *Proganochelys quenstedtii*.

The aim of this work is to revise the material of the Triassic material from Thailand in the light of the current knowledge about the early pantestudinate anatomy. Additionally, more turtle shell fragments from the outcrop of the Huai Hin Lat Formation in Huai Kee Tom in the district of Khonsan, Chaiyaphum Province were also mentioned but not figured or described in detail by Laojumpon *et al.* [35], and newly prepared, previously unpublished material gathered from the type locality in the 1980s was encountered in the collection of the MNHN – these also will be documented herein.

## Geological setting

The Huai Hin Lat Formation of Thailand consists of volcanic rocks and conglomerate in the lower sequence. Pebble conglomerates with fossiliferous limestone clasts dominate [36,37]. The upper sequence consists of calcareous shale and sandstone. Occasional coals are observed. The depositional environment is considered to be of a fluvio-lacustrine type. Temporarily, the Huai Hin Lat Formation extends from the Carnian at the base, to the Norian in the middle and top parts of the sequence [36,38]. Plant fossils are abundant and consists, among others, of *Equisetites*, *Neocalamites*, and *Clathropteris*, and suggest the Norian age for the turtle-bearing strata [38–40]. In addition to turtles, other vertebrate fauna known from the formation includes fish, temnospondyls, such as *Cyclotosaurus* cf. *posthumus* Fraas, 1913 [41], plagiosaurs, and phytosaurs, as well as tetrapod footprints [35,38,42–47]. Other fossils such as freshwater conchostracans, ostracods, and stromatolites are also reported from this formation [48]. Moreover, the outcrops of the Huai Hin Lat Formation yielded numerous coprolites [35,49,50].

## Material

All Triassic turtle material from Thailand (Figs 1, 2A, 3, 4, 5A–C, 6, 7, 8A–D, 9, 10A, 11–13, 14A, B, 15) comes from the Huai Hin Lat Formation (Norian). SM2015-1-001 and

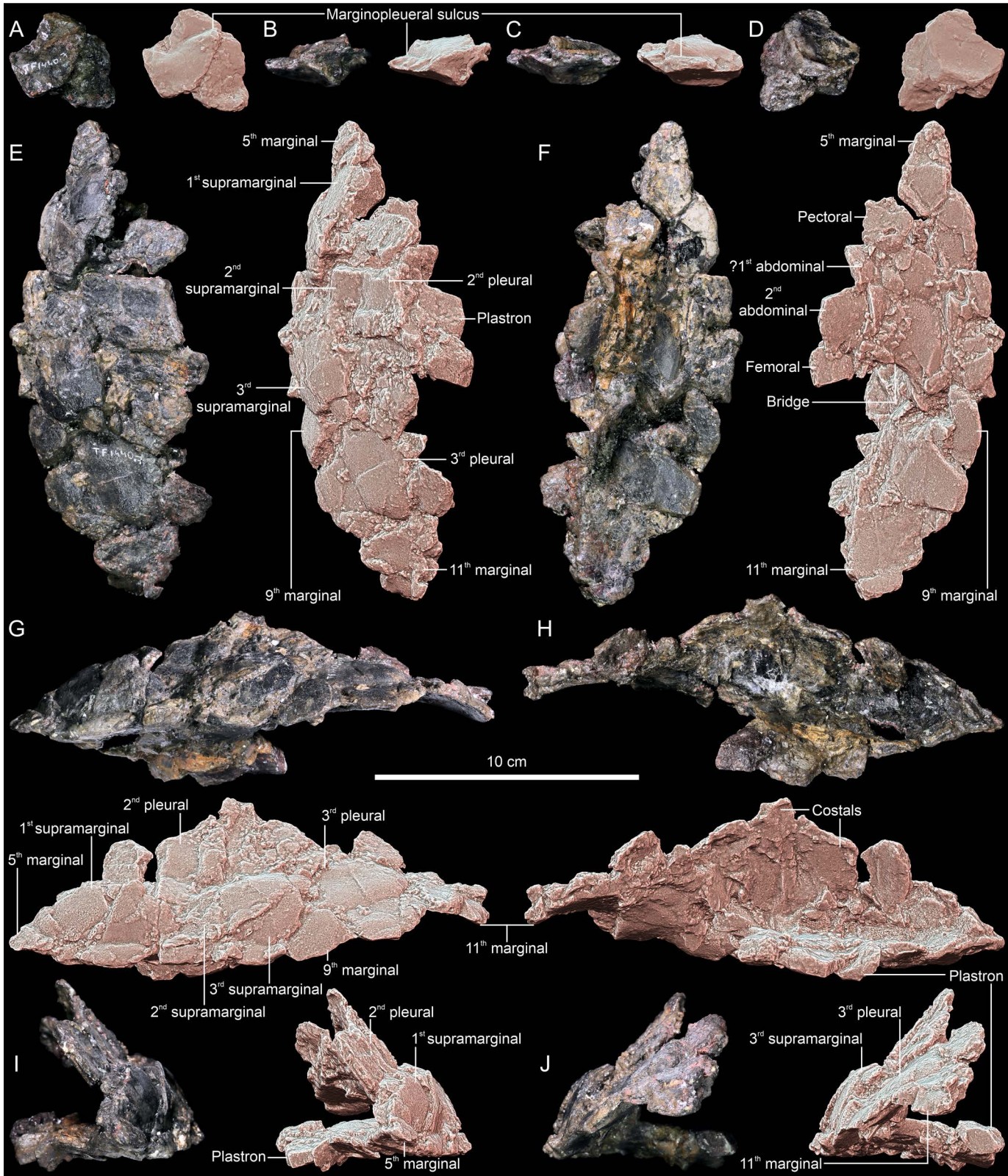

**Fig 1. Thaichelys ruchae, shell fragments. A–D**, SM2017-1-124, part of the anterior left carapace margin in **(A)** dorsal, **(B)** posterior, **(C)** anterolateral (outer), and **(D)** ventral view. **E–J**, SM2017-1-124, left bridge region in **(E)** dorsal, **(F)** ventral, **(G)** lateral, **(H)** medial, **(I)** anterior, and **(J)** posterior view.

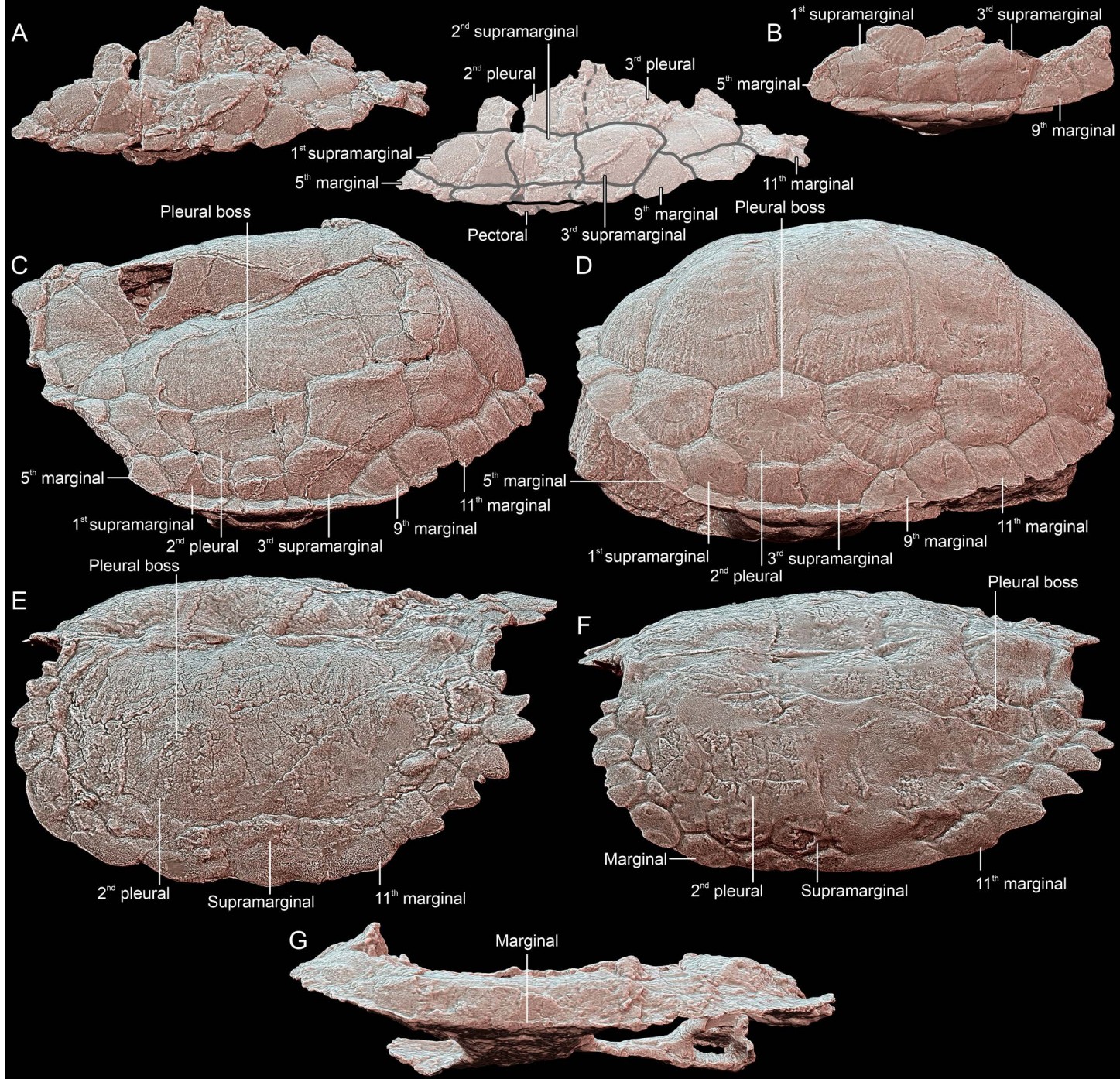

**Fig 2. Comparison of carapace morphology in Triassic turtles (left side, view approximately perpendicular to the supramarginal surface, anterior towards the left, scaled to roughly the same shell length). A**, *Thaichelys ruchae* SM2017-1-124. **B**, *Proterochersis robusta* SMNS 18440. **C**, *Proterochersis porebensis* ZPAL V. 39/48 (holotype). **D**, *Proterochersis robusta* SMNS 17561. **E**, *Proganochelys quenstedtii* MB.1910.45.2. **F**, *Proganochelys quenstedtii* SMNS 16980. **G**, *Palaeochersis talampayensis* PULR 068 (holotype).

SM2017-1-124–SM2017-1-136 (announced without specimen numbers already by Broin et al. [1], described and figured by Broin [2]) were found in 1981 in thick-bedded muddy limestone, conceivably placed on top of basal conglomerate of Huai Hin Lat Late Triassic non-marine sediments, which rests unconformably on top of the Permian limestone. During

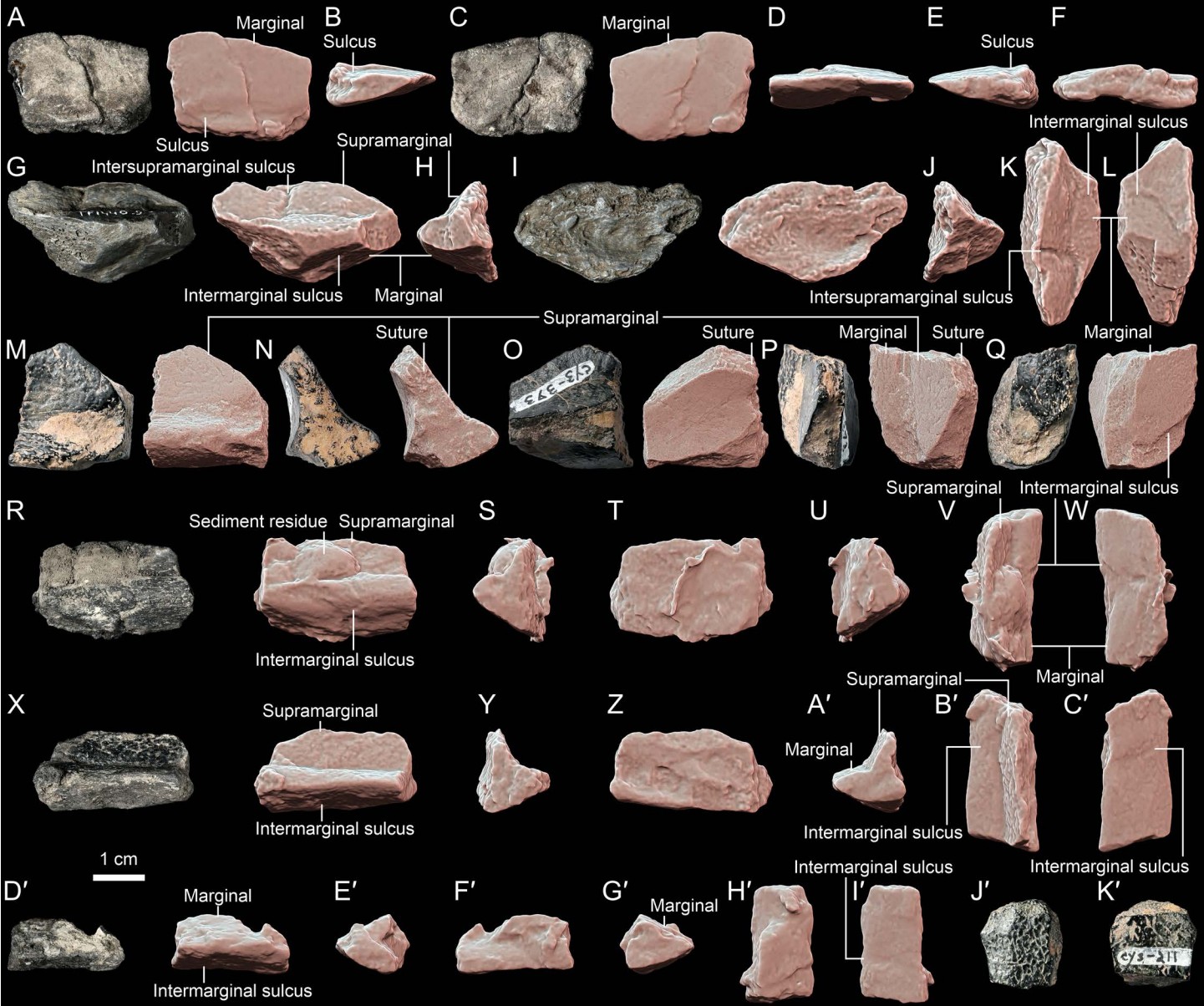

**Fig 3.** *Thaichelys ruchae,* **carapace margin fragments. A–F**, MNHN.F.THA11, part of the presumed anterior left carapace margin in **(A)** dorsal, **(B)** anteromedial, **(C)** ventral, **(D)** outer (anterolateral), **(E)**, posterolateral, and **(F)** basal (medial) view. **G–L**, MNHN.F.THA22.3 (cast of TF 1440-5c), carapace part from the left bridge region in **(G)** external (lateral), **(H)** anterior, **(I)** visceral (medial), **(J)** posterior, **(K)** dorsal, and **(L)** ventral view. **M–Q**, PRC 202, carapace part from the left bridge region in **(M)** external (lateral), **(N)** anterior, **(O)** visceral (medial), **(P)** dorsal, and **(Q)** ventral view. **R–W**, MNHN.F.THA14, carapace part from the right bridge region in **(R)** external (lateral), **(S)** anterior, **(T)** visceral (medial), **(U)** posterior, **(V)** dorsal, and **(W)** ventral view; **X–C′**, MNHN.F.THA13, carapace part from the left bridge region in **(X)** external (lateral), **(Y)** anterior, **(Z)** visceral (medial), **(A′)** posterior, **(B′)** dorsal, and **(C′)** ventral view; **D′–I′**, MNHN.F.THA15, carapace part from the right bridge region in **(D′)** external (lateral), **(E′)** anterior, **(F′)** visceral (medial), **(G′)** posterior, **(H′)** dorsal, and **(I′)** ventral view; **J′**, **K′**, PRC 197, part of the carapace periphery (views not determined).

the course of this study, we were made aware that the MNHN collection houses a number of newly prepared specimens collected at that time from the Ban Suan Sawan Banana Farm, Si Chomphu, Khon Kaen. Although most of those constitute indeterminate bone debris, several are informative. According to France de Lapparent de Broin (pers. com., 2024), blocks of rock are still present in the MNHN collection and may contain additional specimens but still

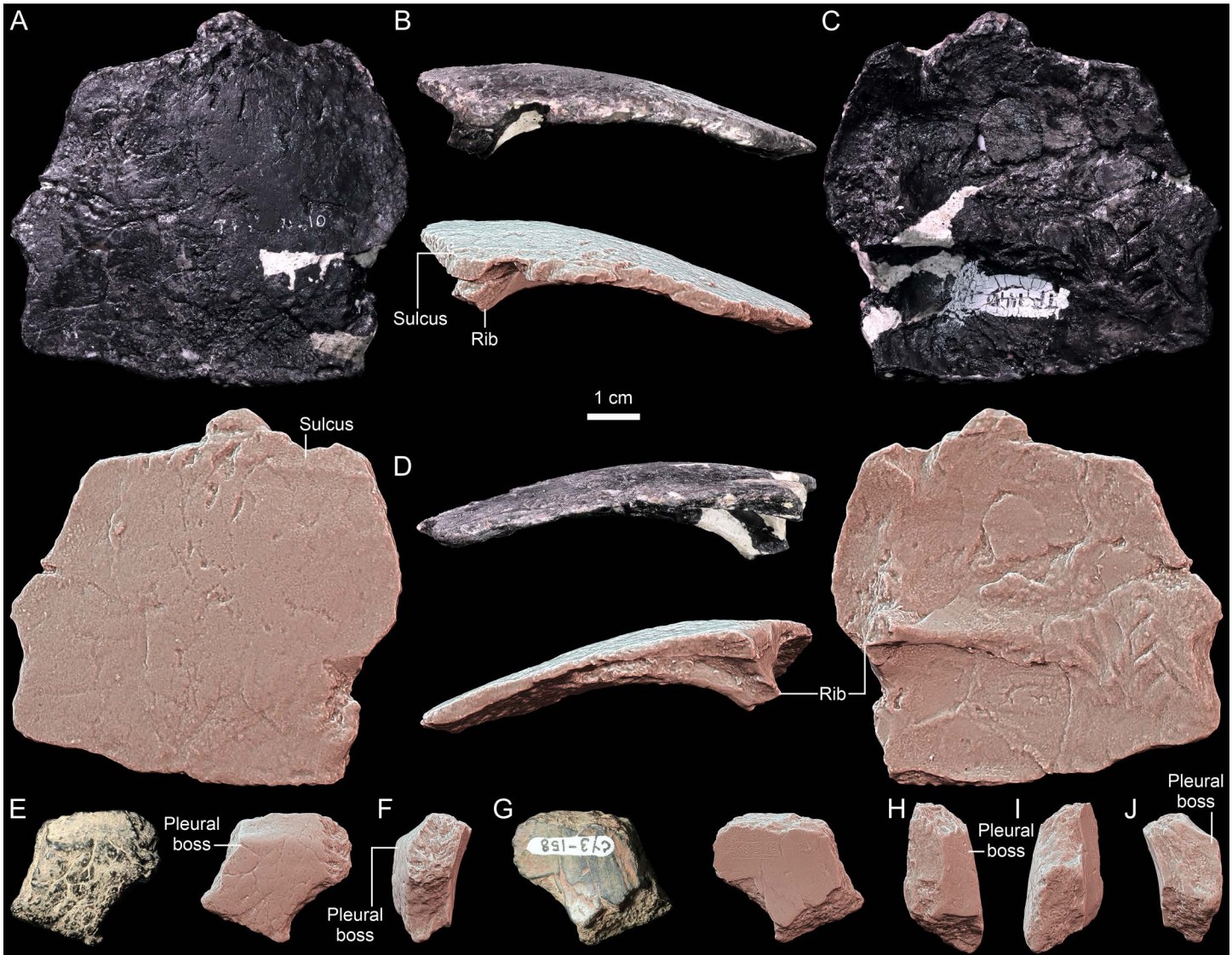

**Fig 4. *Thaichelys ruchae*, carapace fragments.** A–D, SM2017-1-128, dorsomedial part of the carapace (side not determined) with a partial rib shaft in (A) external, (B, D), anterior or posterior, and (C), visceral side. E–J, carapace part with a pleural boss in (E) external, (F, J) anterior or posterior, (G) visceral, (H) dorsal, and (I) ventral view.

await processing. As stated by specimen labels, recognizable turtle material from the Huai Kee Tom in the PRC collection was found in 2010 (PRC 195 – former CY3-158, PRC 196 – former CY3-160, PRC 197 – former CY3-211, PRC 198 – former CY3-212, PRC 199 – former CY3-213, PRC 200 – former CY3-214, and PRC 201 – former CY3-215) and 2011 (PRC 202 – former CY3-373) and, according to Laojumpon *et al.* [35], collected from the exposed surface of the ground.

Material found in 1980 along the road from Ban Huai Sanan Sai to Na Pha Song (TF 1440-5a, TF 1440-5b, and TF 1440-5c), initially described by Broin *et al.* [1] as an indeterminate turtle and later [2] referred to *Thaichelys ruchae* (her *Proganochelys ruchae*), could not be located in the SM collection and is presumed lost. However, good quality casts of those specimens (except of the crushed shell fragment associated with TF 1440-5c and figured by Broin *et al.* [1]: fig 2D) are housed in the MNHN (MNHN.F.THA22.1, MNHN.F.THA22.2,

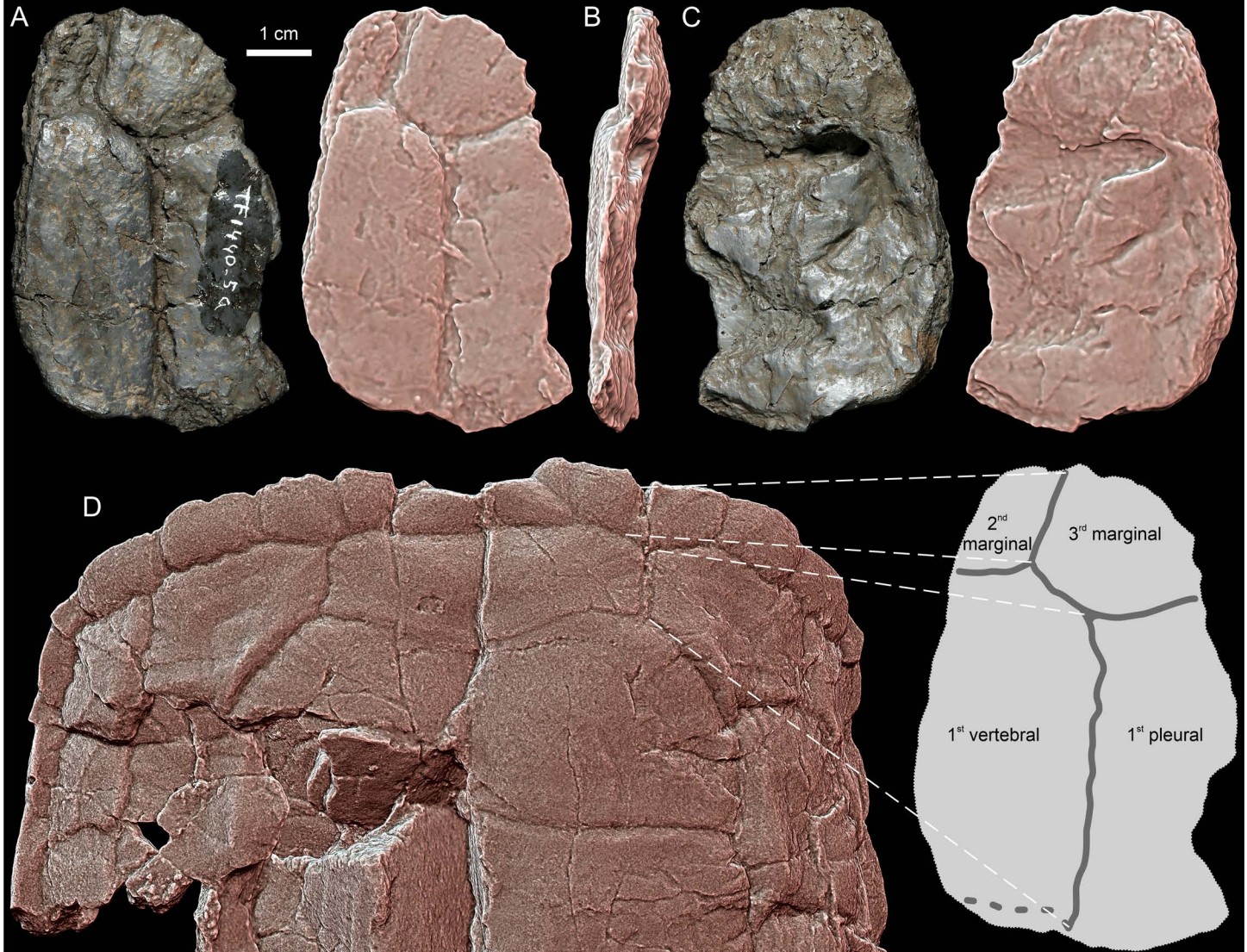

**Fig 5.** *Thaichelys ruchae* **MNHN. F.THA22.1 (cast of TF-1440-5a), presumed anterolateral right part of the carapace. A–C**, specimen in (**A**) external, (**B**) presumed lateral, and (**C**) visceral view. **D**, comparison [1]with the anterior part of the carapace of *Proterochersis porebensis* ZPAL V. 39/49 (paratype, left) in dorsal view. Scute sulci indicated with grey lines, dashed grey lines indicate poorly defined or ambiguous sulci. D not to scale.

MNHN.F.THA22.3) and were examined. TF 1440-5a can be tentatively identified as a part of the anterior right portion of the carapace (partial areas of the second and third marginal scutes, the first vertebral scute, and the first pleural scute). TF 1440-5b is a difficult to identify fragment of the shell. TF 1440-5c comes from the right side and consists of partial areas of two bridge marginals and two supramarginals – its more precise location is uncertain.

SM2015-1-001 (former TF 1440-6, the holotype of *Thaichelys ruchae*) is a 10.7 cm long fragment of the left side of the anterior plastral lobe, preserving the gular and extragular projection, base of the dorsal (ascending) entoplastral process, and a part of the humeral scute area. From the same locality come three more partial anterior plastral lobes: MNHN.F.THA17 (left, consisting of the gular and extragular projection, laterally deflected part of the dorsal epiplastral process, and a smaller than in SM2015-1-001 part of the humeral scute area),

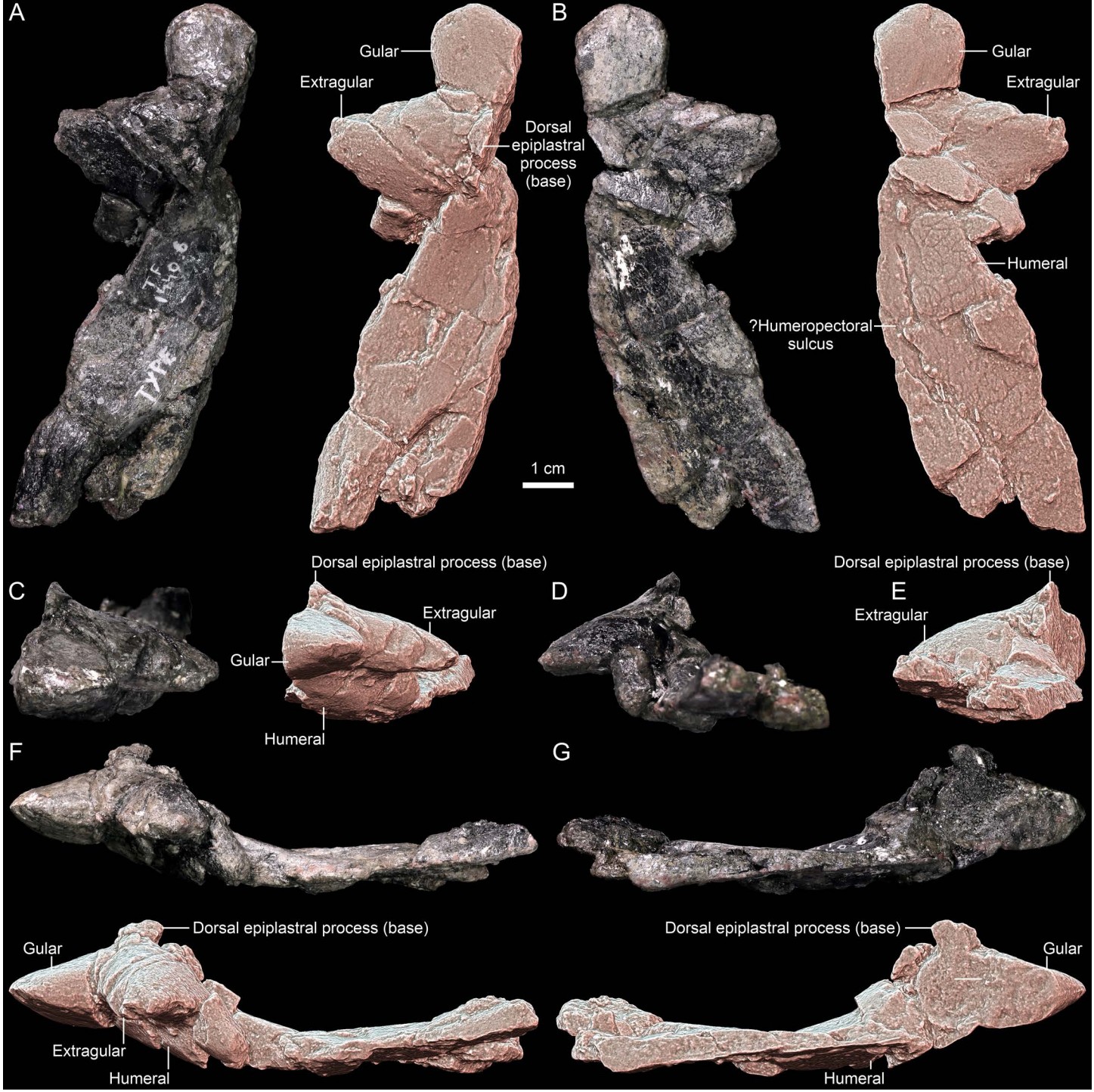

**Fig 6. *Thaichelys ruchae* SM2015-1-001 (holotype), left side of the anterior plastral lobe. A**, dorsal view. **B**, ventral view. **C**, anterior view. **D**, posterolateral view. **E**, posterior view. **F**, lateral view. **G**, medial view.

MNHN.F.THA21 (right, nearly equivalent in the represented area with MNHN.F.THA17 but smaller, preserving even smaller part of the humeral scute area and less of the dorsal epiplastral process, yet undeformed), and MNHN.F.THA18 (right, with broken extragular projection

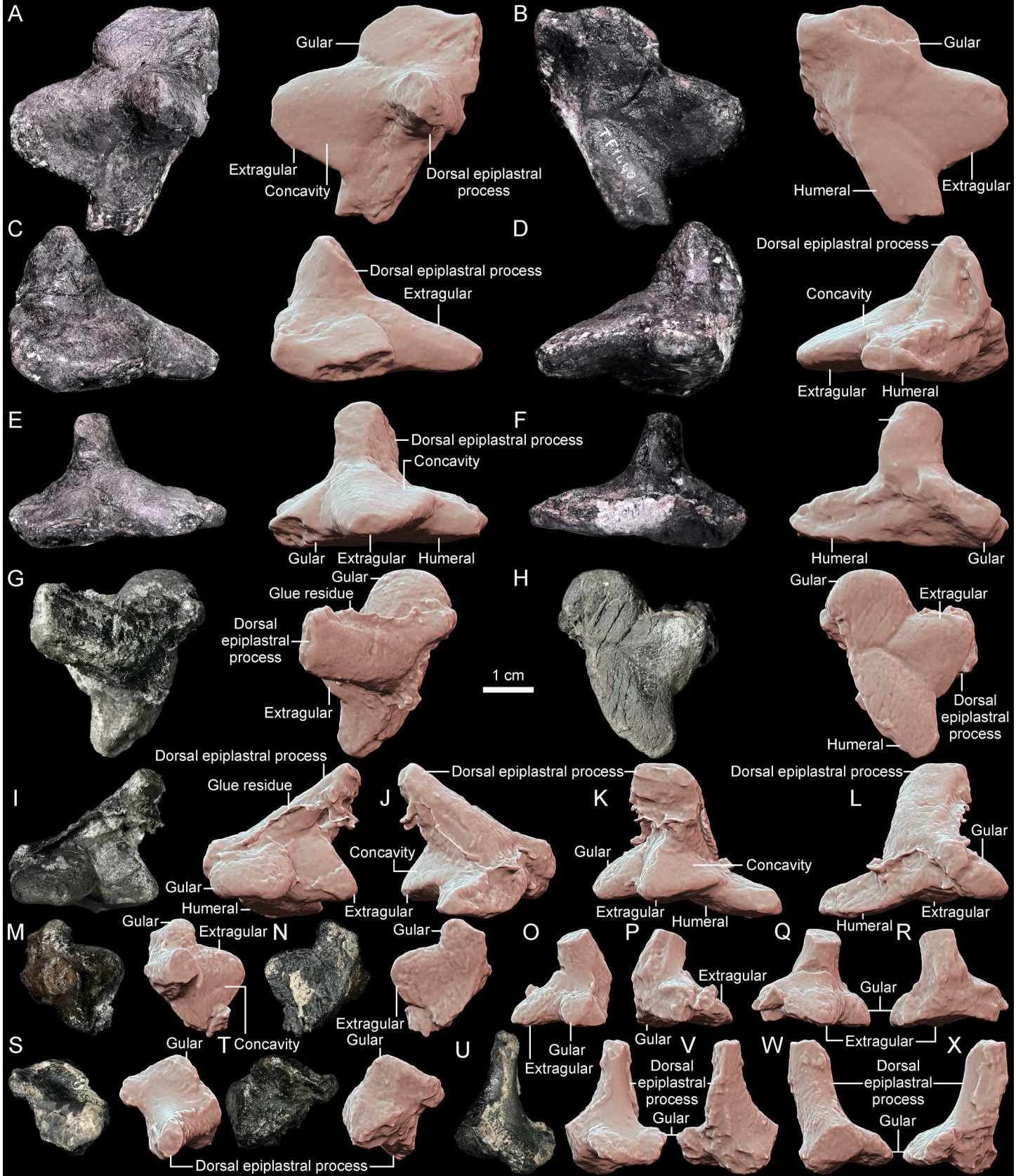

**Fig 7. *Thaichelys ruchae*, partial anterior plastral lobes. A–F**, SM2017-1-129 (left) in **(A)** dorsal, **(B)** ventral, **(C)** anterior, **(D)** posterior, **(E)** lateral, and **(F)** medial view. **G–L**, MNHN.F.THA17 (left) in **(G)** dorsal, **(H)** ventral, **(I)** anterior, **(J)** posterior, **(K)** lateral, and **(L)** medial view. **M–R**, MNHN.F.THA21 (right) in **(M)** dorsal,

(N) ventral, (O) anterior, (P) posterior, (Q) lateral, and (R) medial view. S–X, MNHN.F.THA18 (right) in (S) dorsal, (T) ventral, (U) anterior, (V) posterior, (W) lateral, and (X) medial view.

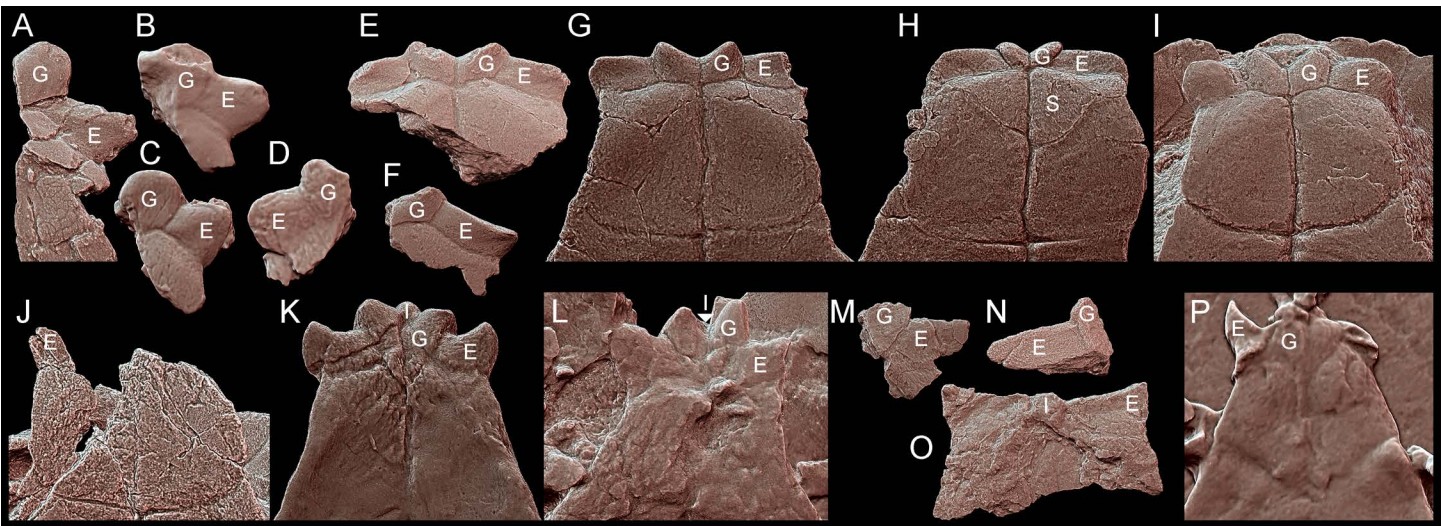

**Fig 8. Comparison of the gular region morphology in Triassic pantestudinates (ventral view, scaled to roughly the same shell length).** **A**, *Thaichelys ruchae* SM2015-1-001 (holotype, left side). **B**, *Thaichelys ruchae* SM2017-1-129 (left side, gular damaged). **C**, *Thaichelys ruchae* MNHN.F.THA17 (left side). **D**, *Thaichelys ruchae* MNHN.F.THA21 (right side, gular damaged). **E**, *Proterochersis porebensis* ZPAL V. 39/420 (left side). **F**, *Proterochersis porebensis* ZPAL V. 39/48 (holotype, both sides). **G**, *Proterochersis porebensis* ZPAL V. 39/49 (paratype, both sides, left extragular damaged). **H**, *Proterochersis porebensis* ZPAL V. 39/385 (both sides). **I**, *Proterochersis robusta* SMNS 17561 (both sides, right extragular restored). **J**, *Keuperotesta limendorsa* SMNS 17757 (part of the right extragular). **K**, *Proganochelys quenstedtii* SMNS 16980 (both sides). **L**, *Proganochelys quenstedtii* SMNS 17204 (both sides, deformed). **M**, *Proganochelys quenstedtii* SMNS 17203 (left side). **N**, *Proganochelys quenstedtii* SMNS 15759 (partial right epiplastron). **O**, *Palaeochersis talampayensis* PULR 068 (holotype, both sides). **P**, *Odontochelys semitestacea* IVPP V 13240 (paratype, both sides, right extragular somewhat dislocated, right extragular missing). **E**: extragular; **G**: gular; **I**: intergular; **S**: supernumerary scute.

but more complete dorsal epiplastral process and seemingly more juvenile features of the gular projection than in MNHN.F.THA21, despite comparable size). All four specimens were found at the Ban Suan Sawan Banana Farm, Si Chomphu, Khon Kaen.

SM2017-1-125 (former TF 1440-7) represents the left posterior part of the bridge area including the lateral parts of the areas of the pectoral and abdominal (probably double) scutes, anterolateral part of the femoral, several marginals (most likely 5–11), presumed three supramarginals, and probably parts of the second and third pleural. The element is badly broken, so the limits of individual scutes are, in places, difficult to trace. Found at the Ban Suan Sawan Banana Farm, Si Chomphu, Khon Kaen. The specimen was apparently associated with SM2017-1-124 (also former TF 1440-7), a bone fragment probably representing the anterolateral left part of the carapace margin. Four more specimens from the same locality represent parts of the carapace margin. MNHN.F.THA11 is restricted to a single marginal area, presumably also anterolateral left, but broken and possibly somewhat deformed. Three are parts of the bridge region: MNHN.F.THA14, from the right side of the body, is roughly comparable in extent to TF 1440-5c but the supramarginal region is obscured by matrix; MNHN.F.THA13 comes from the left side and represents parts of two marginal and one supramarginal scute areas; and MNHN.F.THA15 represents two partial marginal areas from the right side. As in the case of TF 1440-5c, these specimens are too incomplete to determine their exact position. MNHN.F.THA12 is a partial left hyoplastron, and MNHN.F.THA20 is another plastral fragment, possibly the left xiphiplastron, partially obscured by rock residue and with the visceral surface covered in a layer of glue.

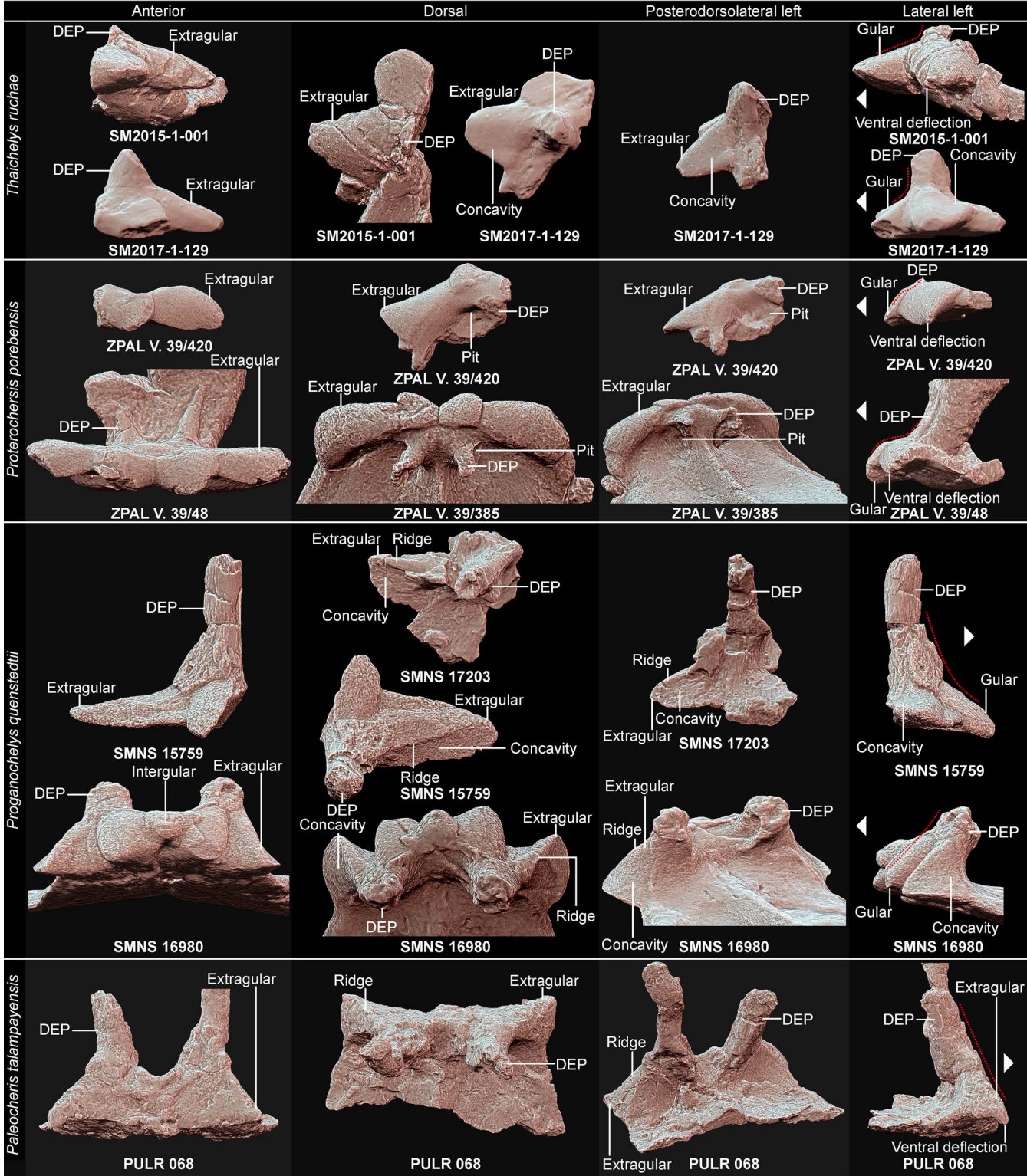

**Fig 9. Comparison of the gular region morphology in Triassic pantestudinates (anterior, dorsal, posterodorsolateral left, and lateral left view, scaled to roughly the same shell length).** *Proterochersis porebensis* ZPAL V. 39/385 mirrored (left to right) in posterolateral view for easier comparison. **DEP**: dorsal epiplastral process. In lateral view, arrowhead marks the anterior direction and red dotted lines indicate the transition between the anterior, scute covered face of the gular and the dorsal epiplastral process.

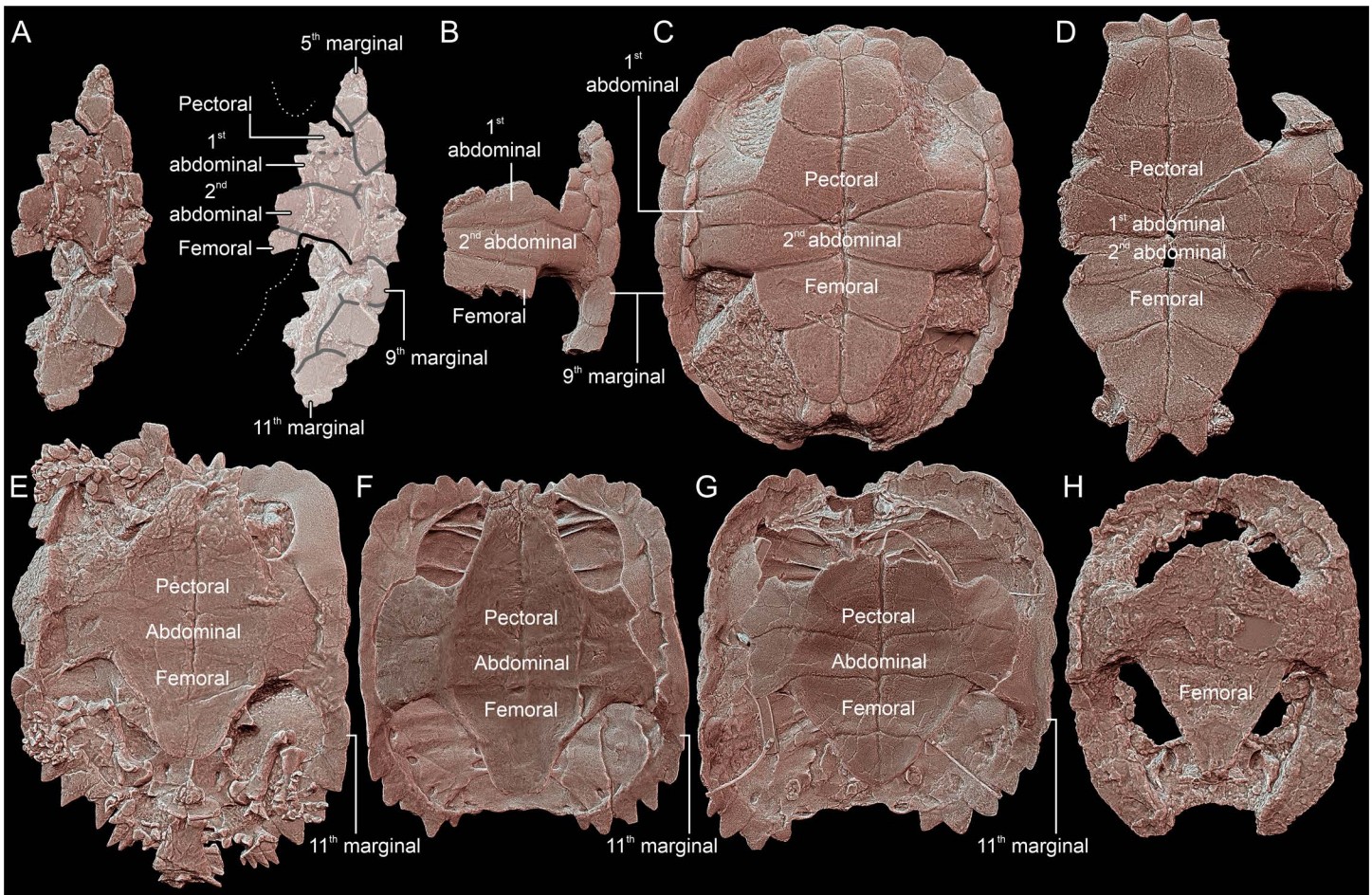

**Fig 10. Comparison of the bridge region morphology in Triassic pantestudinates (ventral view, scaled to roughly the same shell length). A**, *Thaichelys ruchae* SM2017-1-124. **B**, *Proterochersis robusta* SMNS 18440. **C**, *Proterochersis robusta* SNBS 17561. **D**, *Proterochersis porebensis* ZPAL V. 39/49 (pratype). **E**, *Proganochelys quenstedtii* SMNS 17204. **F**, *Proganochelys quenstedtii* SMSN 16980. **G**, *Proganochelys quenstedtii* MB.1910.45.2. **H**, *Palaeochersis talampayensis* PULR 068 (holotype). Scute sulci in **A** indicated with grey lines, dashed lines indicate poorly defined or ambiguous sulci.

SM2017-1-126 (former TF 1440-8) and SM2017-1-127 (former TF 1440-9) are small, irregular indeterminate bone fragments. According to the label, SM2017-1-127 comes from Ban Huai Sanam Sai, Nam Nao, Phetchabun. Broin [2] stated that both specimens come from the route from Chum Phae to Loei, at the Huai Hin Lat stream. No formation is listed in the label but the locality represents the type section of the Huai Hin Lat Formation, according to Broin [2].

MNHN.F.THA10 is a plate from the Ban Suan Sawan Banana Farm, Si Chomphu, Khon Kaen, in some respects resemblant of the elements of the dermal carapacial mosaic of proterochersids. Due to the imperfect preservation, however, this identification must be considered with caution.

SM2017-1-128 (former TF 1440-10) is a carapace fragment representing a part of the vertebral scute area, with a proximal part of the rib preserved viscerally. A short fragment of a transverse, likely intervertebral sulcus is preserved dorsally. SM2017-1-129 (former TF 1440-11) is another left anterior part of the plastron, comparable in the represented area with MNHN.F.THA17, preserving the gular and extragular projection, base of the dorsal (ascending) entoplastral process (more complete than SM2015-1-001), and a small part of

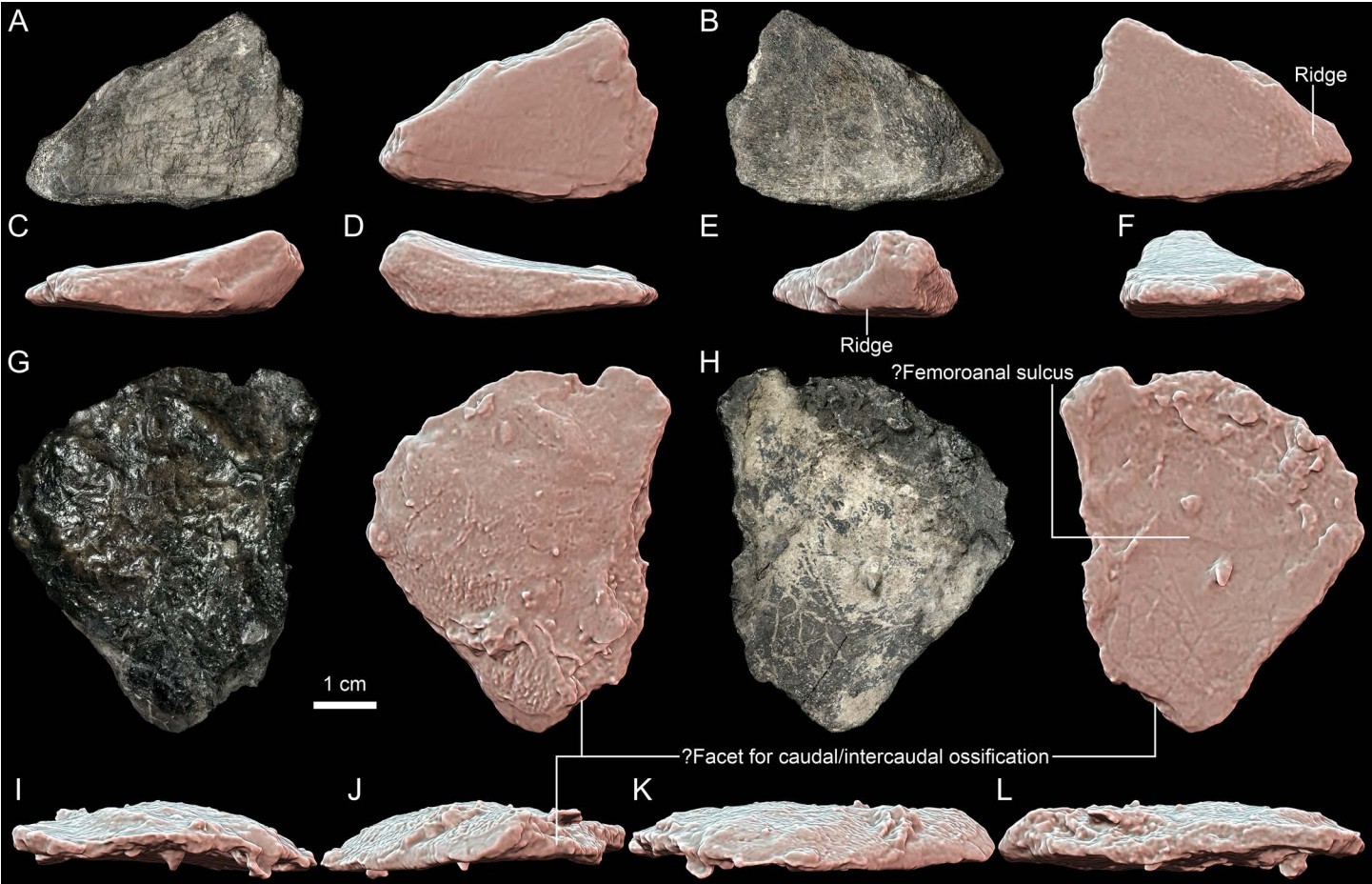

**Fig 11. *Thaichelys ruchae*, plastral fragments. A–F**, MNHN.F.THA12, fragment of the left hyoplastron in **(A)** visceral (dorsal), **(B)** external (ventral), **(C)** anterior, **(D)** posterior, **(E)** lateral, and **(F)** medial view. **G–L**, MNHN.F.THA20, probable left xiphiplastron in **(G)** visceral (dorsal), **(H)** external (ventral), **(I)** anterior, **(J)** posterior, **(K)** lateral, and **(L)** medial view.

the humeral scute area. According to the associated label, it comes from Ban Huai Sanam Sai, Nam Nao, Phetchabun.

MNHN.F.THA16 is a left humerus from the Ban Suan Sawan Banana Farm, Si Chomphu, Khon Kaen. The bone is complete but compacted and partially covered in a mixture of glue and sediment in places projecting as fin-like extensions and obscuring the natural outline of the bone. MNHN.F.THA19 from the same locality is a distal part of a radius, presumably from the right side.

SM2017-1-130 (former TF 1440-12a) is a probable partial left ilium from Ban Huai Sanam Sai, Nam Nao, Phetchabun. Several other, mostly indeterminate bone fragments also coming from Ban Huai Sanam Sai were attributed by Broin [2] to the same taxon: SM2017-1-131 (former TF 1440-12b), SM2017-1-132 (former TF 1440-12c), SM2017-1-133 (former TF 1440-12d), SM2017-1-134 (former TF 1440-12e), SM2017-1-135 (former TF 1440-12f), SM2017-1-136 (former TF 1440-12g). At least one of them (SM2017-1-132), however, appears to be a part of a vertebral arch (possibly atlas) of an animal different than turtle. We tentatively follow Broin's [2] regarding the attribution of the remaining specimens although they present no diagnostic characters.

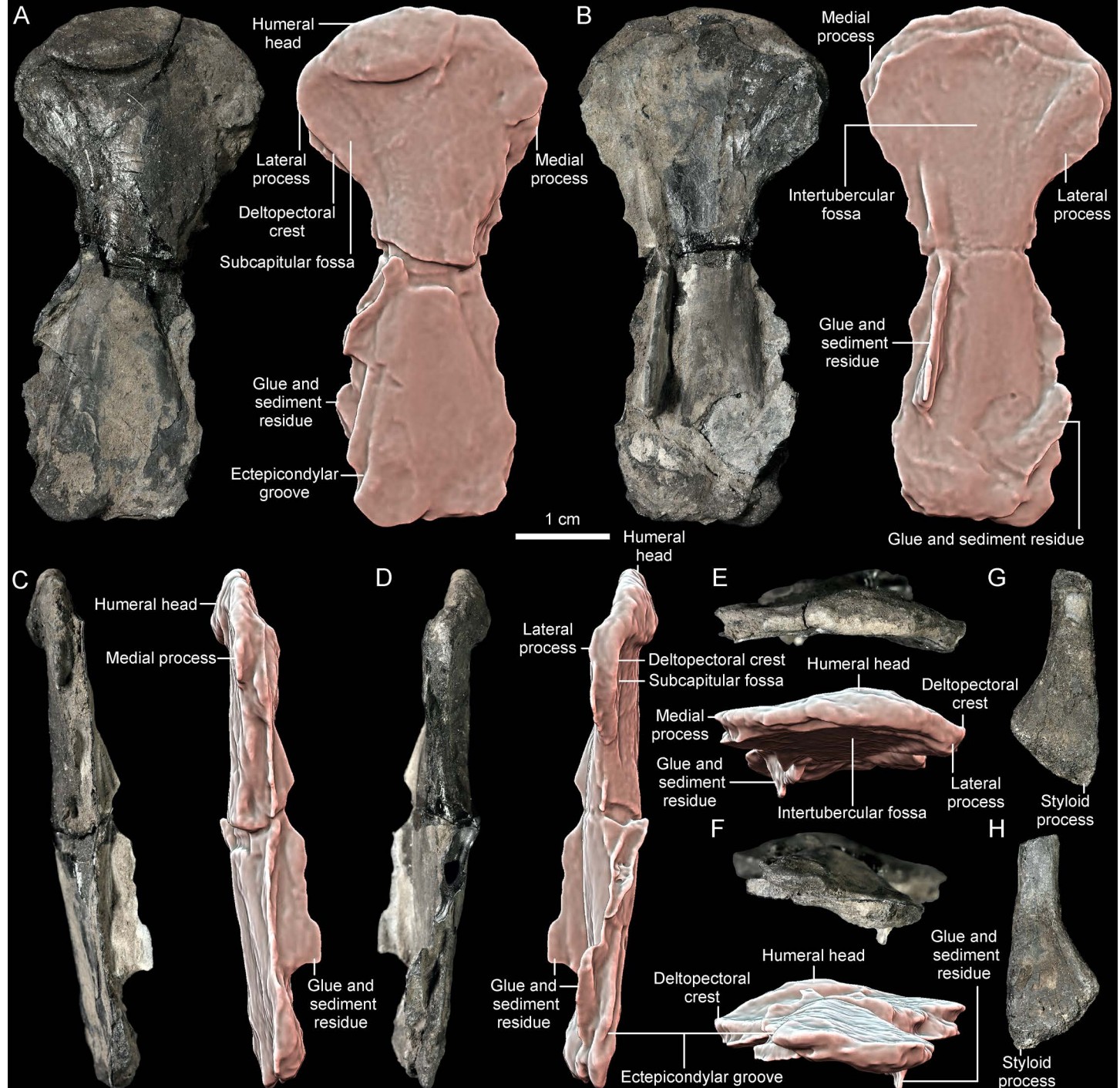

**Fig 12. _Thaichelys ruchae_, forelimb bones. A–F**, MNHN.F.THA16, left humerus in (**A**), capitular, (**B**) intertubercular, (**C**) ulnar, (**D**) radial, (**E**) proximal, and (**F**) distal view. **G, H**, MNHN.F.THA19, (?right) radius, distal part in (**G**)?dorsal and (**H**)?ventral view.

PRC 195, PRC 196, PRC 197, PRC 198, PRC 199, PRC 200, PRC 201, and PRC 202 are shell fragments from Huai Kee Tom. Although their diagnostic value is limited, they show no incongruence with the material described by Broin _et al._ [1] and Broin [2], giving no evidence for a second turtle taxon in the Huai Hin Lat Formation. For that reason, they are tentatively

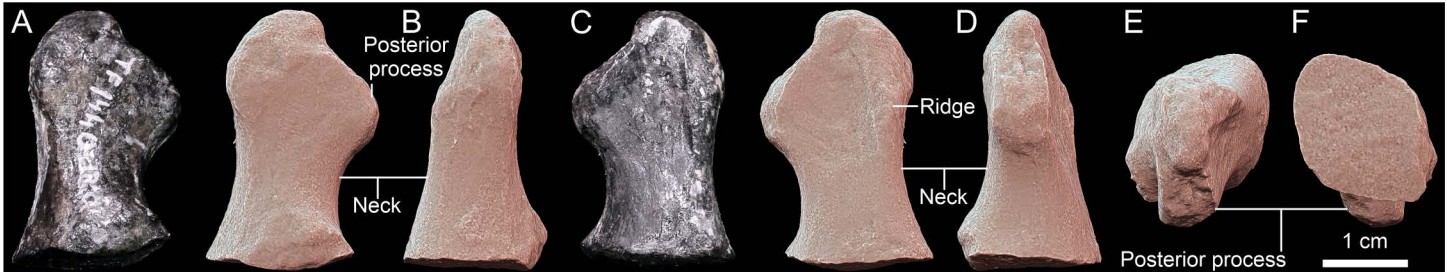

**Fig 13.  *Thaichelys ruchae* SM2017-1-130, partial left ilium. A**, lateral view. **B**, anterior view. **C**, medial view. **D**, posterior view. **E**, dorsal view. **F**, ventral view.

assigned to the same taxon as the material from other localities of the same formation. Aside from the listed specimens, the PRC collection from that locality includes multiple specimens, some of which are recognizable as temnospondyls or archosaurs, and others unidentifiable.

The preservation of the Huai Hin Lat Formation material is unfortunately relatively poor. Although the bones may locally preserve some fine surface detail, almost always the specimens are broken into small fragments, making recognition of their initial shape and scute sulci tricky. The bones are generally black, dark grey, or dark brown and frequently preserve residues of light grey, reddish, or ochre-colored rock matrix. They are reportedly difficult to prepare from the matrix, requiring prolonged acid baths (France de Lapparent de Broin, pers. com., 2024).

## Methods

The material of *Thaichelys ruchae*, *Chinlechelys tenertesta* Joyce, 2009 [8] (NMMNH), *Keuperotesta limendorsa* Szczygielski and Sulej, 2016 [16] (SMNS), *Odontochelys semitestacea* Li, 2008 [14] (IVPP), *Palaeochersis talampayensis* Rougier *et al.*, 1995 [5] (PULR), *Proganochelys quenstedtii* (MB, SMF, SMNS), *Proterochersis porebensis* Szczygielski and Sulej, 2016 [16] (ZPAL), *Proterochersis* cf. *porebensis* (ZPAL), *Proterochersis robusta* Fraas, 1913 [51] (CSMM, NHMUK, SMNS), and *Waluchelys cavitesta* Sterli *et al.*, 2021 [7] (PVSJ) was examined personally by the authors. See Supporting Information for the list of examined specimens. Additionally, Triassic turtle material from Greenland (NHMD) was studied but it will be tackled in a separate contribution.

Shell bone and scute terminology follows Zangerl [52] and Hutchison and Bramble [53]. Directional terminology for the humerus follows Hermanson et al. [54].

### Specimen digitization

Selected specimens were digitized using photogrammetry (*Thaichelys ruchae* in the SM and PRC collections, *Odontochelys semitestacea*, *Palaeochersis talampayensis*, large and complex specimens of other taxa) or surface 3D scanning (*Thaichelys ruchae* in the MNHN collection, smaller separated shell fragments of *Keuperotesta limendorsa*, *Proganochelys quenstedtii*, and *Proterochersis* spp.). The photogrammetry was performed with Agisoft Metashape Professional 2.0.1 (photographs aligned on the High setting, mesh produced from depthmaps on the High or Ultra High setting) and the resulting models were automatically scaled based on printed scale-markers. Surface scanning was performed with a Shining 3D EinScan Pro 2X 3D surface scanner fixed on a tripod with EinScan Pro 2X Color Pack (texture scans), Ein-Turntable (alignment based on features), and EXScan Pro 3.7.0.3 and 3.7.4.0 software. The number of turntable steps was varied, chosen depending on the fragment, and

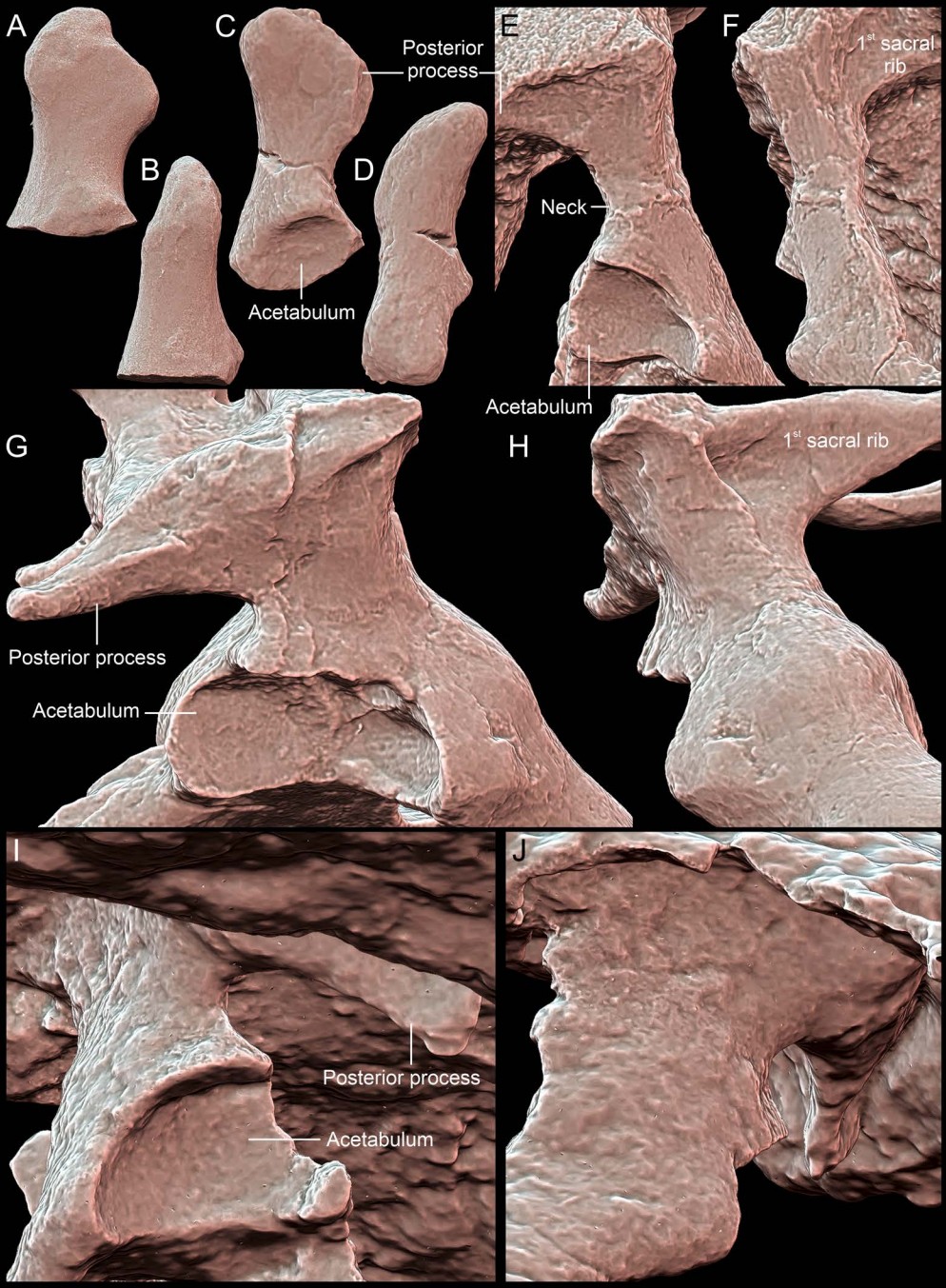

**Fig 14. Comparison of the ilium morphology in Triassic pantestudinates (scaled to roughly the same ilium length).**
**A, B**, *Thaichelys ruchae* SM2017-1-130, partial left ilium in (**A**) lateral and (**B**) anterior view. **C, D**, *Proterochersis porebensis* ZPAL V. 39/177, partial left ilium in (**C**) lateral and (**D**) anterior view. **E, F**, *Proterochersis robusta* SMNS 56606, right ilium in (**E**) lateral and (**F**) anterior view. **G, H**, *Proganochelys quenstedtii* SMNS 16980, right ilium in (**G**) lateral and (**H**) anterior view. **I, J**, *Palaeochersis talampayensis* PULR 068 (holotype) in (**I**) lateral and (**J**) anterior view.

the resulting models were meshed using the Watertight Model and High Detail presets. To better visualize the morphology, snapshots of 3D models were captured in MeshLab 2021.10 [55] in orthographic view and with the Radiance Scaling (Lambertian, Lit Sphere, and Grey

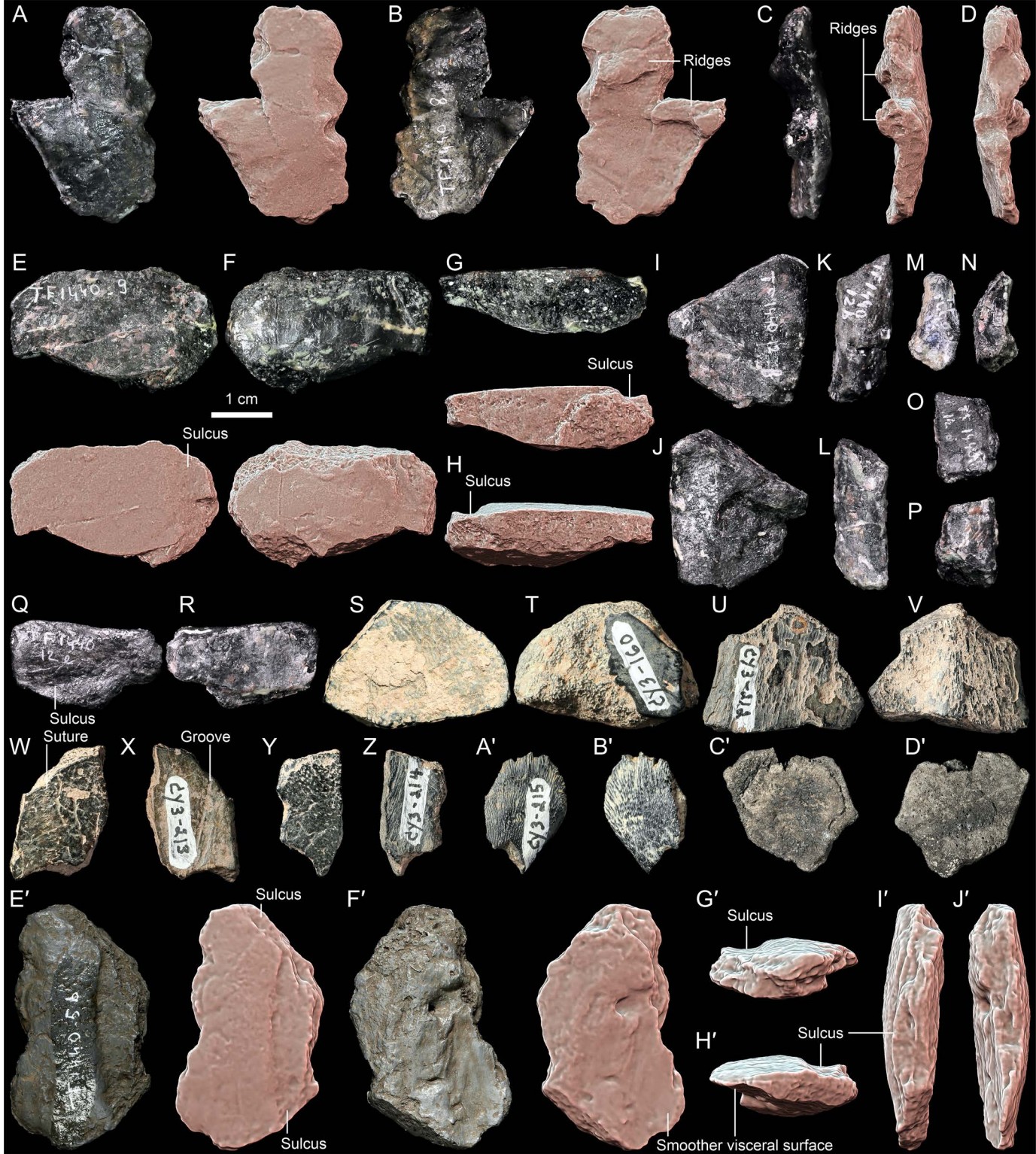

**Fig 15. *Thaichelys ruchae*, anatomically indeterminate material. A–D**, SM2017-1-126, shell fragment in **(A)** external, **(B)** visceral, and **(C, D)** edge views. **E–H**, SM2017-1-127, shell fragment in **(E)** external, **(F)** visceral, and **(G, H)** edge views; **I, J**, SM2017-1-135, shell fragment in **(I)** external and **(J)** visceral view. **K, L**, SM2017-1-131, appendicular bone shaft or part of the dorsal epiplastral process or dorsal scapular process. **M, N**, SM2017-1-136, appendicular bone shaft or part of the dorsal epiplastral process. **O, P**, SM2017-1-133, shell fragment. **Q, R**, SM2017-1-134, carapace periphery or plastron fragment. **S, T**, PRC 196,

shell fragment in (**S**) external and (**T**) visceral view. **U, V,** PRC 198, shell fragment. **W, X,** PRC 199, probable fragment of two sutured costals or plastral bones in (**W**) external and (**X**) visceral view. **Y, Z,** PRC 200, shell fragment in (**Y**) external and (**Z**) visceral view. **A′, B′,** probable fragment of juvenile plastron. **C′, D′,** MNHN.F.THA10 polygonal plate (?part of carapacial dermal mosaic) in (**C′**) external and (**D′**) visceral view. **E′–J′,** MNHN.F.THA22.2 (cast of TF 1440-5b), shell (?carapace) fragment in (**E′**) external, (**F′**) visceral, and (**G′, H′, I′, J′**) edge view. Precise orientation of all specimens not determined.

Descriptor) shader enabled [56]. All the generated 3D models are accessible for download from MorphoSource: https://www.morphosource.org/projects/000683358.

## Phylogenetic analysis

The phylogenetic analysis utilized the matrix of and Sterli *et al.* [7], based on earlier iterations of the matrix created by Sterli *et al.* [57] and subsequently used and updated by López-Conde *et al.* [58], Sterli *et al.* [59], and Sterli and de la Fuente [60], with two taxa (*Chinlechelys tenertesta* and *Thaichelys ruchae*) and 22 new characters added, resulting in a total of 106 taxa and 276 characters. The scorings for new taxa and corrections to the scorings for *Keuperotesta limendorsa*, *Mongolemys elegans* Khozatsky & Młynarski, 1971 [61], *Odontochelys semitestacea*, *Palaeochersis talampayensis*, *Plesiochelys etalloni* Pictet and Humbert, 1857 [62], *Proganochelys quenstedtii*, *Proterochersis* spp., and *Stylemys nebrascensis* Leidy, 1851 [63] were made based on personal examination of relevant specimens. See Supporting Information for the list of characters, specimens and references used for rescorings and scoring new characters, matrix, and lists of synapomorphies. The analysis was performed in TNT 1.6 [64,65] with *Odontochelys semitestacea* as the outgroup and using New Technology Search (Sectorial Search, Ratchet, Drift, and Tree fusing enabled) with the Find minimum length option set to 30 times. Bootstrap values were calculated for the topology of the strict consensus with 1000 replications.

**Removed character.**   Extragular C, anterior plastral tuberosities (0) present, (1) absent. We had difficulty replicating this character. Among stem turtles, *Proganochelys quenstedtii* and *Proterochersis* spp. were scored as 0 but the Australochelyidae Gaffney and Kitching, 1994 [66] were scored as 1, despite the morphology of their extragulars being relatively similar to that observed in some specimens of *Proterochersis porebensis* [6,7,18] and the morphology in *Proganochelys quenstedtii* being variable but arguably more distinct [2,4]. All other turtles (not counting taxa for which this character is unapplicable or its state is unknown) were scored as 1, except *Otwayemys cunicularius* Gaffney *et al.*, 1998 [67] and *Chelus fimbriatus* (Schneider, 1783) [68] (both scored as 0), in which the extragulars are gently rounded anterolaterally [67], different than in Triassic taxa and more similar to some specimens of *Plesiochelys etalloni* [69], which was scored as 1. Because of that, and because we added several new characters concerning the morphology of the gulars and extragulars, this character is removed.

**Modified characters.**   Abdominal A. State 2 (absent) is now removed due to the introduction of Abdominal B. Taxa scored here for 2 are rescored to?. *Proterochersis robusta* is here scored as 0 (present, with medial contact), because the morphology in SMNS 17561 with the first pair of abdominals separated in the middle and the second pair meeting at the midline (Fig 10C) is considered abnormal (it is not present in any other specimen of the species; e.g., Fig 10B).

Dorsal rib B. "Dorsal rib 9–10" redefined to "the last two dorsal rib pairs", given the larger number of dorsal ribs in Triassic pantestudinates [7,15–17].

Dorsal rib C. "Dorsal rib X" redefined to "last dorsal rib" in both states, given the larger number of dorsal ribs in Triassic pantestudinates [7,15–17].

Extragular D. State 0 (only on the epiplastra) was redefined to "not reaching the entoplastron". This is because at least in some Triassic stem turtles the lateral part of the extragular scutes was supported by an additional extragular ossification, separate from the epiplastra

[19,28], which may not be detectable in other species due to frequent shell ankylosis. State 1 remains unchanged (reach the entoplastron).

**New characters.** Abdominal B, (0) two pairs, (1) one pair, (2) absent. Ordered. The number, medial contact and presence of abdominal scutes are apparently at least partly independent, given the morphology in *Proterochersis robusta* SMNS 17561, in which the first pair is separated in the middle and the second pair meets at the midline (Fig 10C). While we consider this morphology abnormal, it is symmetric and shows that likely there was a developmental capacity in early turtles to independently control the medial extent of the abdominals. Moreover, while the exclusion of the abdominal scutes by their medial separation and subsequent decrease of their area is a plausible scenario, we see no reason to assume that their loss by a gradual decrease in length along their whole width or simply early fusion or loss of scute placodes are implausible. Therefore, we conclude that the number of abdominal scutes (2, 1, 0) constitutes a separate morphocline possibly independent from their medial extent, and thus it is scored here as a separate character.

Cervical vertebra L, (0) eighth presacral vertebra co-ossified with the carapace and succeeding vertebrae, (1) eighth presacral vertebra has an intermediate (transitional) cervico-dorsal morphology, can be sutured to the carapace but not to the succeeding vertebrae, (2) eighth presacral vertebra free from the carapace and succeeding vertebrae, movable. Ordered. Modified from Szczygielski & Sulej [16] following the interpretations of Szczygielski [17].

Coracoid, (0) flat, sub-ovoid or with rounded posteromedial edge (bee-wing shaped), (1) flat, rectangular or with a distinctly angular posteromedial edge, (2) columnar, at least at its base. Modified from Szczygielski & Sulej [16].

Costal E, (0) 10 pairs, (1) 9 pairs, (2) 8 pairs or less. Ordered. Taken from Szczygielski & Sulej [16]. Anteriormost and posteriormost dorsal ribs in turtles tend to be reduced in size. The first dorsal rib pair may in such cases either remain free from the carapace (e.g., Triassic taxa) or attach suturally to the first costal plate or the underlying shaft of the second dorsal rib pair. The posterior ribs may, likewise, attach to the preceding costals, rib shafts, or ilia, or be free. Therefore, the common number of costals in recent turtles is eighth. However, stem turtles frequently have more costals due to the contribution of those anterior or posterior dorsal rib pairs to the carapace and/or an increased number of dorsal segments, and the increased number of costal is sometimes variably present in extant turtles. Because the number of costals and dorsal ribs are not inextricably correlated, these characters are treated here separately.

Dorsal epiplastral process A, (0) lateroventroposterior excavation present, resulting in a distinct depression lateral to the anterior part of the medial ridge on the visceral surface of the entoplastron, partially roofed dorsally by the base of the dorsal epiplastral process, (1) lateroventroposterior excavation absent, no dorsally roofed depression in that area. The morphology of the dorsal epiplastral processes exhibits a significant variability in Triassic taxa (Fig 9) but was thus far not considered in detail. The topological features concerning the geometry of the anterior plastral lobe and probable muscle and ligament attachments may be phylogenetically significant and thus are treated here. This character is concerned with the presence of a distinct, pocket-like fossa anteriorly walled by the anterior lip of the gular and extragular edge, medially walled by the raised midline ridge formed by the entoplastron, and dorsally partially roofed by the base of the dorsal process of the epiplastron, possibly accommodating the distal end of the acromion or the acromial ligament [19,28]. Thus far this structure is only documented in *Proterochersis* spp. (Fig 9). Despite the overall similarity of that region between *Thaichelys ruchae* and *Proterochersis* spp., it is absent in the former.

Dorsal epiplastral process B, (0) posterior relative to the body wall, separated from the rugose, scute-covered areas of the gular and extragular projections by a band of smooth bone and a flexure, (1) abutting the body wall, the anterior surface of the dorsal epiplastral process

lies in the same plane and constitutes a continuation of the scute-covered surface of the extragular projection. The position of the dorsal process of the epiplastron varies between Triassic taxa, in some of them (e.g., *Proterochersis* spp., *Thaichelys ruchae*) it is clearly separated from the gular and extragular scutes by a distinct flexure and a band of smoother bone while in *Proganochelys quenstedtii* and the australochelyids there is no clear border and the dorsal process continues in the same plane as the anterodorsal surface of the gulars and/or extragulars (Fig 9).

Dorsal epiplastral process C, (0) base distinctly broadened ventrally in anterior view, triangular, forming an extension of the ascending dorsolateral edge of the extragular projection, (1) base narrow, clearly separated from the dorsolateral edge of the extragular projection by a distinct difference in the angle between these structures. The bases of the dorsal epiplastral processes in *Proterochersis* spp. and *Thaichelys ruchae* are clearly separated from the extragular projections not only by their relative position but also due to their very abrupt deflection [28]. In *Proganochelys quenstedtii* and the australochelyids this transition is much smoother (Fig 9) [2,4,6,7].

Dorsal epiplastral process D, (0) located closer to the midline than to the lateral edge of the anterior plastral lobe, not extending significantly beyond the lateral extent of the gular scutes, (1) located more laterally, about halfway or more between the midline and the lateral edge of the anterior plastral lobe. In *Eorhynchochelys sinensis* and *Odontochelys semitestacea* the dorsal processes of the clavicles are located far laterally [14,15], in *Proganochelys quenstedtii* and the australochelyids they take an intermediate position [4,6,7] and are positioned closer to the midline in *Thaichelys ruchae*, *Proterochersis* spp., and *Keuperotesta limendorsa* (Fig 9) [2,16,19,28].

Dorsal rib E, (0) more than 10 pairs, (1) 10 pairs, (2) 9 pairs or less. Ordered. Modified from Szczygielski & Sulej [16]. This character specifies the number of segments in the trunk. Based on personal observation of the available material, *Odontochelys semitestacea* and *Proterochersis* spp. have an additional dorsal segment compared to more derived turtles, i.e., eleven rather than ten dorsal vertebrae and rib pairs [16,17]. Eleven pairs of dorsal ribs were also reported for *Waluchelys cavitesta* by Sterli *et al.* [7] and twelve pairs were reported for *Eorhynchochelys sinensis* Li *et al.* [15], establishing likely character polarity within the early Mesozoic pantestudinates. *Eunotosaurus africanus* is stated to have nine dorsal segments [70]. Posterior dorsal ribs are absent in kinosternids without the reduction of dorsal vertebrae [71]. Sacralization of the posterior dorsal vertebrae and ribs occurs in Pleurodira, and while developmentally the affected ribs differ from the true sacral ribs [72], they are functionally excluded from the dorsal region. Because the number of costals and dorsal ribs are not inextricably correlated, these characters are treated here separately.

Extragular edge A, (0) anterior edge convex, straight, or nearly straight, (1) anterior edge distinctly concave in a horn-like fashion. The gular and extragular region of the plastron is one of the most distinct parts of the shell in Triassic turtles, especially considering its relative massiveness and relatively good preservation compared to other parts of the shell in some species, but thus far was only cursorily tackled in previous phylogenies. It is particularly significant in the case of *Thaichelys ruchae*, therefore, a number of characters are here introduced concerning the morphology of the relevant structures. Note that due to improvements in character sampling the value of these new characters may improve in the future. The extragular projections are straight in most taxa but curved anteriorly in *Odontochelys semitestacea* and some specimens of *Proganochelys quenstedtii* (Figs 8–10) [4,14,73].

Extragular edge B, (0) lateral or anterolateral tip spiky, (1) lateral or anterolateral tip rounded. The tip of the extragular projections is variably spiky or rounded in *Proterochersis porebensis* but the spiky variant is rare (Figs 8E–H 8) [16,18]. Variability in that regard has not been detected thus far in other taxa.

Extragular edge C, (0) ventral deflection of the anterior edge present, the process concave ventrally and comma shaped in cross-section, (1) ventral deflection of the anterior edge absent, the process flat or convex ventrally and V- or U-shaped in cross-section. In some specimens of *Proterochersis porebensis*, *Thaichelys ruchae*, and the australochelyids, the anterior edge of the extragular process is deflected downwards, resulting in a convex ventral surface of the projection and a lip along the anterior edge (Fig 9) [6,7,16,18,19]. The degree of that deflection can vary [18]. The deflection is absent in *Odontochelys semitestacea*, *Proterochersis robusta*, and *Proganochelys quenstedtii* [4,14].

Extragular process A, (0) large, projected about third or more of the extragular scute area, (1) minor, the anterolateral edge of the anterior plastral lobe gently scalloped, (2) none, the anterolateral edge of the anterior plastral lobe even. Ordered. This character is concerned with the size of the extragular projection and is substituted for the removed character Extragular C.

Extragular process B, (0) dorsal, anterolaterally or laterally directed ridge extending from the tip of the extragular process to the base of the dorsal epiplastral process present and distinct, (1) absent or weak. In *Odontochelys semitestacea* (IVPP V 15639), *Proganochelys quenstedtii*, and the australochelyids, there is a distinct ridge connecting the tip of the extragular projection to the dorsal process of the epiplastron (Fig 9) [4,6,7]. This ridge is absent in *Thaichelys ruchae*, *Proterochersis* spp., and *Keuperotesta limendorsa* [2,19].

Extragular process C, (0) dorsal surface of the projection convex, (1) dorsal surface distinctly concave in the posterolateral part of the projection. This character refers to the dorsal, scute covered edge of the extragular. In *Proganochelys quenstedtii* the posterolateral facet of the extragular process (posterior to the ridge described by the Extragular process B character) is characteristically depressed (Fig 9) [4]. A very subtle depression was apparently present in some specimens of *Thaichelys ruchae* but it is not known in other taxa. In many modern taxa the dorsal exposure of the extragular scute is very restricted but, as long as it is present and shows no posterolateral depression, it is scored as 1. Not to be confused with the posteromedial depression related to the body wall attachment.

Femur A, (0) articular surface of femoral head triangular in dorsal view, (1) rectangular or oval. Modified from Szczygielski & Sulej [16].

Gular edge, (0) spiky, distinctly conical, (1) rounded at the end to straight, (2) spiky, minor part of the scute width contributing to the spike. As evidenced by *Proterochersis porebensis* and *Thaichelys ruchae*, the shape of the gular and extragular projections is regulated independently (Figs 8A–H , 8) [2,18], therefore the gulars are coded separately from the extragulars.

Gular process, (0) large, projected about third or more of the gular scute area, (1) minor, the anterior edge of the anterior plastral lobe gently scalloped, (2) none, the anterolateral edge of the anterior plastral lobe even. Ordered.

Marginal B, (0) serration of posterior marginal scutes in adults pronounced, rounded tips of underlying peripherals, (1) serration of posterior marginal scutes in adults pronounced, spiky tips of underlying peripherals, (2) weak or no serration of posterior marginal scutes in adults. Taken from Szczygielski & Sulej [16]. This character does not consider the notch between the posteriormost marginals, if this is the only notch present in the posterior edge of the carapace.

Sacrum, (0) contact of the neural spines of sacral vertebrae with carapace ossified, (1) chondral, ligamentous or none. Modified from Szczygielski & Sulej [16]. This character currently unites only the species of *Proterochersis* spp. [16,19,28], its distribution in some other Triassic taxa is unknown but may prove phylogenetically valuable in the future.

Scapulocoracoid, (0) angle between coracoid and acromion 130 degrees or less, (1) more than 130 degrees. Modified from Szczygielski & Sulej [16].

Vertebral E, (0) first vertebral scute bell-shaped, wide anteriorly and tapering posteriorly into a narrower, rounded median process invading the area of the wide second vertebral scute, (1) first vertebral scute subrectangular, hexagonal, or trapezoid with posterior edge roughly transverse and not significantly narrower than the anterior edge of the second vertebral scute, (2) first vertebral scute bell-shaped but does not invade the area of the second vertebral scute. *Proterochersis* spp. and *Keuperotesta limendorsa* are distinguished from other turtles by the unusual shape of their first vertebral scute [16,18,22]. A similar morphology seems to be also present in *Chinlechelys tenertesta* – figured but not noticed by Lucas and Lichtig ([9]: fig 9), confirmed by personal observation of NMMNH P-16697. Outside of that group, roughly the same morphology of the first vertebral scute is also present in *Sichuanchelys* spp. [74–76]. In some turtles (e.g., *Xenochelys formosa* Hay, 1906 [77]) the first vertebral scute approaches the bell shape but it ends posteriorly in a transverse sulcus and does not invade the area of the second vertebral scute. In those latter cases the posteriorly narrowed shape of the first vertebral scute seems to be forced by the significant narrowing of the second vertebral scute and the intervertebral sulcus with retention of normal connections between the scutes, so we consider this state non-homologous with that observed in the proterochersids.

## Nomenclatural acts

The electronic edition of this article conforms to the requirements of the amended International Code of Zoological Nomenclature, and hence the new names contained herein are available under that Code from the electronic edition of this article. This published work and the nomenclatural acts it contains have been registered in ZooBank, the online registration system for the ICZN. The ZooBank LSIDs (Life Science Identifiers) can be resolved and the associated information viewed through any standard web browser by appending the LSID to the prefix "http://zoobank.org/". The LSID for this publication is: urn:lsid:zoobank. org:pub:8F027887-AF6E-42F1-85CF-5932DDCCF58A. The electronic edition of this work was published in a journal with an ISSN, and has been archived and is available from the following digital repositories: LOCKSS.

## Results

### Systematic paleontology

*AMNIOTA* Haeckel, 1866 [78] *sensu* Laurin and Reisz, 2020 [79]

*TESTUDINATA* Klein, 1760 [80] *sensu* ITNC *et al.*, 2020a [81]

*PROTEROCHERSIDAE* Nopcsa, 1923 [82] (converted clade name)
Registration number: 985 [created, not submitted yet].
Definition: The largest clade containing *Proterochersis robusta* Fraas, 1913 [51] but not *Proganochelys quenstedtii* Baur, 1887 [3], *Australochelys africanus* Gaffney and Kitching, 1994 [66], and *Testudines* Batsch, 1788 [83] *sensu* ITNC *et al.*, 2020b [84]. This is a maximum-clade definition. Abbreviated definition: max ∇ (*Proterochersis robusta*, 1913 ~ (*Proganochelys quenstedtii* Baur, 1887 & *Testudines* Batsch, 1788)).
Reference phylogeny: Presented herein (Fig 16).
Composition: In addition to *Proterochersis robusta* and *Proterochersis porebensis*, in the reference phylogeny the clade *Proterochersidae* includes *Keuperotesta limendorsa*, *Thaichelys ruchae*, and *Chinlechelys tenertesta*.
Synonyms: Archaeochelydae Fraas, 1913 [51] (amended as Archaeochelyidae by Kuhn, 1960 [85]), including *Proterochersis* but not *Proganochelys quenstedtii* and more derived turtles. Fraas [51] argued that classification of the newly established genus *Proterochersis*

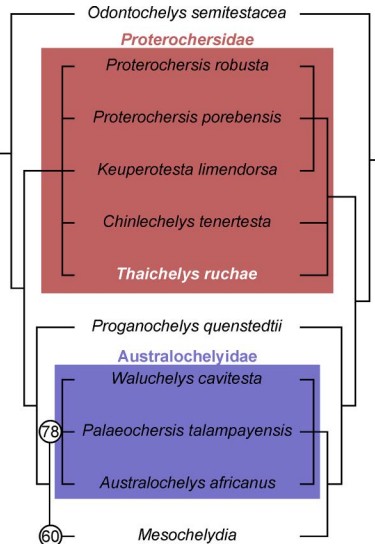

**Fig 16. Strict (left) and Majority Rule (50%, right) consensus trees of the recovered phylogeny.** Bootstrap values above 50 indicated by the numbers in circles.

into a separate family is justified by its unique combination of characters, being a mix of primitive features and derived terrestrial adaptations. The name was never used outside of the Fraas' [51] original paper and checklists by Kuhn [85–87], who acknowledged its priority over the Proterochersidae but still preferred the latter name. Moreover, although Fraas [51] apparently coined his Archaeochelydae independently and did not derive it from any generic name, 'Archaeochelys' Lydekker, 1889 [88] was already preoccupied at the time. To add to the confusion, Bergounioux [88] described a probable septarian nodule [34,89,90] from Permian sediments of France as a turtle shell and named it 'Archaeochelys pougeti', and subsequently erected for it a new family, archaeochelyidés ( = Archaeochelyidae Bergounioux, 1955 [91]; incorrectly listed as Bergounioux, 1938 [92], even though the name was not used in the latter paper). Kuhn [87], convinced that the specimen is indeed a turtle fossil, renamed it to 'Archaeochelydium pougeti' [87,93,94]) and established a new family name to accommodate it: Archaeochelydiidae Kuhn, 1961 [87].

Comments: The family-level name Proterochersidae was first used by Nopcsa in 1923 [82]. Monotypic Proterochersidae was subsequently recognized by Bergounioux [95,96]. Kuhn [86] included in the family both the genus *Proterochersis* (at the time including two species – *Proterochersis robusta* and *Proterochersis 'intermedia'* Fraas, 1913 [51]) and '*Chelytherium obscurum*' Meyer [97] – a problematic genus and species based on fragmentary remains from the Triassic of Germany [97,98], recently found to be a senior synonym of *Proterochersis robusta* [20], but suggested for suppression due to its convoluted taxonomic history (ICZN case 3840), following the explicitly stated preference of earlier authors [24,34]. Nopcsa [99], alongside the genus *Proterochersis*, included also *Saurodesmus robertsoni* Seeley, 1891 [100] – an enigmatic stylopodial bone from the Triassic of Scotland which was initially classified as a probable turtle [101–107] (and even synonymized with '*Chelytherium obscurum*' [88]) or a crocodilian [100], but in fact represents neither of these groups [34,108,109]. The posthumous 1932 edition of Zittel [110], Romer [111], and Bergounioux [91] included in the Proterochersidae the genus *Proterochersis*, '*Chelytherium obscurum*', *Saurodesmus robertsoni*, and the genus '*Chelyzoon*' Huene, 1902 [112] (probably misinterpreted tanystropheid vertebrae [113]). Huene [114] reiterated the same contents, except '*Chelytherium obscurum*', which he

considered synonymous with *Proganochelys quenstedtii*. Kuhn [94,115] returned to the mono-typic idea of the Proterochersidae by removing from that family all taxa but *Proterochersis* spp. Karl [116] included in the Proterochersidae two taxa, *Proterochersis robusta* and '*Murrhardtia staeschei*' Karl and Tichy, 2000 [22]. Szczygielski and Sulej [16] synonymized both *Proterochersis* '*intermedia*' and '*Murrhardtia staeschei*' with *Proterochersis robusta* but erected a new species, *Proterochersis porebensis*, and another new, closely related proterochersid taxon, *Keuperotesta limendorsa*. Joyce [34] proposed synonymy of the genera *Keuperotesta* and *Proterochersis*, resulting once again in the monotypy of the Proterochersidae. While this approach is not invalid, we follow here the reasoning of Parker [117] and Szczygielski and Sulej [19], and prefer to uphold the name *Keuperotesta* to preserve the taxonomic stability and for its utility to differentiate the closely related yet distinct and less complete taxon. Szczygielski and Sulej [19] recovered *Proterochersis* spp. in a single clade with *Chinlechelys tenertesta* but due to the incompleteness of the latter and *Keuperotesta limendorsa* not being a part of the analysis, they favored a more inclusive and conservative composition of the Proterochersidae. Sterli *et al.* [7] once again recovered the monophyletic Proterochersidae consisting of *Proterochersis* spp. and *Keuperotesta limendorsa* but did not include *Chinlechelys tenertesta* in their analysis. Herein, we include all the taxa previously recovered within the clade and *Thaichelys ruchae* (never before included in a cladistic phylogeny), and establish the redefined, more inclusive Proterochersidae as a converted clade name.

*THAICHELYS* Szczygielski et al. gen. nov

urn:lsid:zoobank.org:act:21F2FC59-2638-45E0-8704-3943174115D3

Etymology: Referring to Thailand, as the origin place of the described specimens, and incorporating -*chelys* (gr. Χέλυς, turtle): „Thai turtle".

Type and only species: *Thaichelys ruchae* (Broin, 1984) [2].

Diagnosis: As for the type species (see below).

*Thaichelys ruchae* (Broin, 1984) [2] comb. nov.

Holotype: SM2015-1-001 (former TF 1440-6), incomplete left side of the anterior plastral lobe (Figs 6, 8A, 9).
Type locality: Ban Suan Sawan Banana Farm, Si Chomphu, Khon Kaen, Thailand.
Type horizon and age: Huai Hin Lat Formation, Late Triassic (Norian).
Revised diagnosis: Testudinate characterized by a unique combination of characters: (1) carapace not reduced in thickness (unlike in *Chinlechelys tenertesta*, *Palaeochersis talampayensis*, and *Waluchelys cavitesta*); (2) pleurals with shelf-like, anteroposteriorly elongated bosses (unlike in *Proganochelys quenstedtii*); (3) three supramarginals present (unlike in *Palaeochersis talampayensis*, *Proganochelys quenstedtii*, and *Waluchelys cavitesta*); (4) bridge-level marginals (a) not wedging between the supramarginals (unlike in *Proganochelys quenstedtii*), (b) with anterolaterally projected, V-shaped in lateral view intermarginal sulci (unlike in *Proganochelys quenstedtii*), (c) with the dorsal exposition smaller than the ventral (unlike in *Proganochelys quenstedtii*), but (d) completely separating dorsomedially located scutes from the edge of the carapace (unlike in *Palaeochersis talampayensis* and *Waluchelys cavitesta*); (5) gular projections large (at least about one third of the scute projected beyond the edge of the plastron) and rounded (unlike in *Palaeochersis talampayensis*, *Proganochelys quenstedtii*, *Waluchelys cavitesta*); (6) gularohumeral sulcus nearly transverse, clearly anterior relative to the posterior extent of the extragular scutes (unlike in *Proganochelys quenstedtii*); (7) extragular projections spiky, laterally directed and large, with no dorsal ridge and minor or absent

dorsoposterolateral concavity (unlike *Proganochelys quenstedtii*); (8) dorsal processes of epiplastra located entirely or mostly within the mediolateral extent of the gular scutes (unlike in *Palaeochersis talampayensis*, *Proganochelys quenstedtii*, and *Waluchelys cavitesta*) on an elevation, separated from the anterodorsal surface of the gular scutes (unlike in *Palaeochersis talampayensis*, *Proganochelys quenstedtii*, *Waluchelys cavitesta*), not excavated ventroposterolaterally (unlike in *Proterochersis* spp. and *Keuperotesta limendorsa*); (9) moderately developed inguinal buttress (unlike in *Palaeochersis talampayensis*, *Proganochelys quenstedtii*, and *Waluchelys cavitesta*); (10) posterolateral inclination of the posterior edge of the bridge (unlike in *Proterochersis* spp. and *Keuperotesta limendorsa*); (11) two pairs of abdominal scutes present (unlike in *Proganochelys quenstedtii*); (12) shell with a less coarse ornamentation than in *Proganochelys quenstedtii*, but more than in *Chinlechelys tenertesta*; (13) ilium with a slender neck (unlike in *Palaeochersis talampayensis*, *Proganochelys quenstedtii*, and *Waluchelys cavitesta*, but thicker than in *Chinlechelys tenertesta*, *Proterochersis* spp., and *Keuperotesta limendorsa*).

Referred specimens: SM2017-1-124 (Fig 1A–D), TF 1440-5c (lost; cast MNHN.F.THA22.1 in Fig 3G–L), PRC 197 (Fig 3J′, K′), MNHN.F.THA11 (Fig 3A–F), carapace edge fragments; MNHN.F.THA14 (Fig 3R–W), MNHN.F.THA15 (Fig 3D′–I′), right bridge fragments; PRC 202 (Fig 3M–Q), SM2017-1-125 (Figs 1E–J, 2A, 10A), MNHN.F.THA13 (Fig 3X–C′), left bridge fragments; SM2017-1-126 (Fig 15A–D), SM2017-1-127 (Fig 15E–H), SM2017-1-131 (Fig 15K, L), SM2017-1-133 (Fig 15O, P), SM2017-1-134 (Fig 15Q, R), SM2017-1-135 (Fig 15I, J), SM2017-1-136 (Fig 15M, N), PRC 196 (Fig 15S, T), PRC 198 (Fig 15U, V), PRC 200 (Fig 15Y, Z), PRC 201 (Fig 15A′, B′), TF 1440-5b (lost; cast MNHN.F.THA22.2 in Fig 15E′–J′), shell fragments; SM2017-1-128 (Fig 4A–D), TF 1440-5a (lost; cast MNHN.F.THA22.1 in Fig 4A–C), PRC 195 (Fig 4E–J), PRC 199 (Fig 15W, X), MNHN.F.THA10 (Fig 15C′, D′); carapace fragments; MNHN.F.THA21 (Figs 7M–R, 8D), MNHN.F.THA18 (Fig 7S–X), right parts of the anterior plastral lobe; SM2017-1-129 (Fig 7A–F, 8B, 9), MNHN.F.THA17 (Figs 7G–L, 8C), left parts of the anterior plastral lobeFig; MNHN.F.THA12 (Fig 11A–F), MNHN.F.THA20 (Fig 11G–L), plastron fragments; MNHN.F.THA16 (Fig 12A–F), left humerus; MNHN.F.THA19 (Fig 12G, H), distal part of the (?right) radius; SM2017-1-130 (Figs 13, 14A, B), left ilium. All specimens from the Huai Hin Lat Formation of Thailand.

## Description

**Carapace.**  The carapace is represented by several specimens (SM2017-1-124, SM2017-1-125, SM2017-1-128, PRC 195, PRC 197, PRC 199, PRC 200, MNHN.F.THA11, MNHN.F.THA14, MNHN.F.THA13, MNHN.F.THA15, MNHN.F.THA10, TF 1440-5a, TF 1440-5c), SM2017-1-125 being the most informative. It is not exceptionally thin, in contrast to *Chinlechelys tenertesta* and the australochelyids (*Palaeochersis talampayensis* and *Waluchelys cavitesta*).

SM2017-1-124 (Fig 1A–D) consists of a single bone fragment apparently associated with SM2017-1-125. The specimen appears to preserve no natural edges and only two natural surfaces set at an acute angle. One of these surfaces (considered here ventral) is nearly flat, with minor concavity, while the other (considered dorsal) consists of two convex fields separated by a transverse, gently sinuous sulcus (Fig 1A–C). The element is interpreted as a part of the carapace margin, most likely from the anterolateral part of the left side, with the sulcus possibly representing the contact between the first pleural and fourth marginal. Therefore, it is just anterior to SM2017-1-125. Marginals 3–8 had a clearly larger ventral than dorsal extent in *Proterochersis* spp. and *Keuperotesta limendorsa* (Figs 2B–D, 9B, C) [16,18] while in *Proganochelys quenstedtii* the dorsal and ventral extent was roughly comparable in size in all marginals (Figs 2E, F, 10E–G) [4,73]. MNHN.F.THA11 (Fig 3A–F) is another presumed part

of the anterolateral left edge of the carapace. The specimen preserves only the area of a single marginal scute, with one, gently bowed natural edge and nearly parallel to it, sinuous (probably pleuromarginal) sulcus crossing the presumed dorsal surface. No sulcus is visible ventrally, indicating a larger dorsal than ventral exposition of the scute. Intermarginal sulci are not preserved but the distance between them had to be at least twice as large as between the pleuromarginal sulcus and the free edge. The specimen shows no particular dorsoventral curvature and is relatively shallow dorsoventrally, but that may be an effect of compaction. The specimen is obliquely partitioned by a break and both parts are pushed out of alignment and seemingly slightly deformed plastically. Overall, the shape and proportions of the scute best fit the fourth left marginal of proterochersids [16,18], although the possibility that the element comes from the posterior part of the carapacial rim cannot be completely ruled out.

SM2017-1-125 (Figs 1E–J, 2A, 10A) is interpreted here as the posterior portion of the left bridge region, in agreement with the previous assessment of Broin [2]. The specimen is crushed and severely damaged, with multiple breaks crossing the surface and some fragments dislocated from their original position. This makes reliable tracing of scute sulci difficult. However, we note several characters consistent with the proterochersid affinity of *Thaichelys ruchae*.

The outlines of pleural scutes are poorly preserved in SM2017-1-125 but it seems that represented are partial areas of the second and third left pleural (Figs 1G, 2A). The pleurosupramarginal sulcus is undulating, as typical for Triassic turtles [4,18,73].

We agree with Broin [2] that the supramarginals were present in SM2017-1-125. Their number and shape are ambiguous due to poor preservation and damage, but upon close examination their configuration seems to agree with that observed in *Proterochersis* spp.: three supramarginals, the first and third larger and pentagonal (with, respectively, a distinct anterior and posterior tip), and the second (very poorly preserved in SM2017-1-125) smaller and square (Figs 1E, G, 2A–D) [4,16,18,19,22]. In all known specimens of *Proganochelys quenstedtii* the circumsupramarginal sulci above the bridge are poorly preserved, but in that region they retain a relatively uniform size and pentagonal shape, with a lateroventrally (marginally) directed tip (Fig 2E, F) [4,33,118]. Note that the outline of these scutes presented by Gaffney ([4]: fig 69B, C [118];: fig 3B, C) for *Proganochelys quenstedtii* MB.1910.45.2 is not entirely accurate – while this specimen has the pentagonal shape of the supramarginals relatively poorly indicated (possibly in part due to the damage and partial restoration of that region), they were pentagonal rather than rectangular (compare with Fig 2E and Gaffney [4]: fig 74). At least in *Waluchelys cavitesta*, the supramarginal row was not continuous in the bridge region [7], differentiating that turtle from SM2017-1-125. Poor surface preservation in *Palaeochersis talampayensis* does not allow reliable identification of the supramarginal and pleural positions in the bridge area (Fig 2G) [5,6].

We interpret the marginal scute areas preserved in SM2017-1-125 to most likely correspond to marginals 5–11 (not 7–13, *contra* Broin [2]). The scutes 6–8 were elongated and narrow in the dorsal and lateral aspect (Figs 1E–J, 2A). Their intermarginal sulci seem to roughly coincide with the intersupramarginal sulci but are inclined anterolaterally, so they form a loose herringbone-like pattern when viewed from the direction of their free edges (Figs 1G, 2A). Such an anterior inclination of the bridge-level marginal sulci, both dorsally and ventrally, is less distinct, particularly dorsally, in *Proganochelys quenstedtii* (Fig 2E, F), but common in *Proterochersis* spp. (Fig 2B–D) [4,16,18,19,22,73]. The marginals did not wedge between the supramarginals. As a result, the marginosupramarginal sulci are roughly straight, like in *Proterochersis* spp. (Fig 2B–D) [16,18,19] but in contrast to the pentagonal (wedged dorsomedially between the supramarginals) bridge-region marginals of *Proganochelys quenstedtii* (Fig 2E, F) [4,33,73]. In the australochelyids, in contrast to SM2017-1-125,

the bridge-region marginals had in places a nearly completely reduced dorsal exposition, allowing the pleurals or supramarginals to reach the edge of the carapace (Fig 2G) [6,7]. The probable ninth marginal of SM2017-1-125 was pentagonal and apparently significantly wider posteriorly then anteriorly, due to the wedging out of the third (last) supramarginal, and the subsequent marginals retained this reliably broader form (Figs 1E, G, 2A). This, as well, is consistent with the morphology observed in *Proterochersis* spp. (Fig 2B–D) [16,18–20], although the widening of the marginals posterior to the inguinal notch is also shared with *Proganochelys quenstedtii* and the australochelyids (Fig 2E–G) [4,6,7,33,73]. However, in *Proganochelys quenstedtii* this also associated with increasing spikiness and serration of the posterior marginals (Figs 2E, F, 10E–G) [4,33,73,119], which is not evident in SM2017-1-125. Despite minor damage, the preserved edge of the carapace in the latter specimen is largely intact and shows only slight serration and rounded rather than spike-like free edges (Figs 1E–H, 2A). The observed serration is somewhat stronger than in *Proterochersis* spp. (Figs 2B–D, 10C) [16,18,19] but much weaker than in *Proganochelys quenstedtii* (Figs 2E, F, 10E–H) [4,33,73,119,120].

The original specimen TF 1440-5c [1] could not be located and is presumed lost, but its cast (MNHN.F.THA22.3) can be interpreted as a part of the carapace rim from the right bridge area (Fig 3G–L). It consists of the partial areas of two marginal and two supramarginal scutes. The visceral surface is concave and uneven. The specimen has a V-shaped outline in the anteroposterior aspect, indicating that it connected the rest of the carapace to the plastron. The thin lappet of bone, interpreted by Broin et al. [1] as the dorsal part of the specimen ending in a natural edge, was most probably directed ventrally (based on the shape and proportion of marginal scutes) and at least in the cast shows no sutural characteristics, indicating that it was broken rather than natural. The intermarginal sulcus is distinct ventrally, where it has a clear anterolateral inclination, and less indicated and located close to the broken edge dorsally. The marginals had at least two times as large exposition ventrally than dorsally. In contrast to SM2017-1-125, the intermarginal sulcus is positioned clearly anterior to the intersupramarginal sulcus. Although in *Proterochersis* spp. these sulci are usually roughly in line with each other (Fig 2B–D), in several specimens (e.g., ZPAL V. 39/49, ZPAL V. 39/72) they exhibit some anteroposterior misalignment (Fig 5D) [16,18,19]. However, there was no wedging of marginals between the supramarginals.

PRC 202 is another part of the carapace periphery from the left bridge region (Fig 3M–Q). The specimen preserves parts of two marginal and probable one supramarginal scute areas. The anterior, posterior, and dorsomedial edges of the presumed supramarginal scute area are not preserved, making it difficult to precisely identify the location of the specimen along the bridge. In contrast to SM2017-1-125 and TF 1440-5c, the presumed marginosupramarginal sulcus of PRC 202 has raised edges (Fig 3M, N, P). This difference, however, fits within the normal variability of scute sulci observed in Triassic turtles [4,18] and thus is not considered to be taxonomically important. Otherwise, the morphology of both specimens is congruent, including the anterolateral inclination of the intermarginal sulci and larger exposition of the marginals ventrally than dorsally. The specimen preserves sutural surfaces anteriorly (interperipheral, Fig 3N) and anterodorsally (part of the costoperipheral suture, Fig 3P). The latter is located relatively high, in the presumed supramarginal area, similarly as in *Proterochersis robusta* but higher than in *Proterochersis porebensis* [19,121]. The ventromedial flange of the bridge is broken off rather than separated along a non-sutured scar (Fig 3N, O), suggesting (but not unambiguously confirming) an osseous connection between the plastron and carapace, rather than ligamentous one. The break occurred laterally to the ventromedial edge of the marginal, so the inframarginal area is not preserved (Fig 3Q), disallowing confirmation of the presence of inframarginal scutes.

SM2017-1-128 (Fig 4A–D) was identified by Broin [2] as a partial costal, and we agree with that identification. The morphology and size of the underlying rib shaft (Fig 4B–D) is similar as in the small *Proterocheris porebensis* ZPAL V. 39/2 [18]. The bone is thickened at the point of rib shaft into the costal, forming an anteriorly and posteriorly fading ridge. The shaft is broken off at the neck and there is no neural process, nor the rest of the vertebra preserved (Fig 4C). Externally, a short portion of a shallow transverse intervertebral sulcus with gently raised edge is visible (Fig 4A). The side of the body is not determined. The rib shaft is slightly slanted relative to the costal plate which may indicate that it may either come from the anterior left or posterior right part of the carapace.

PRC 195 is an element of a carapace from the region of a pleural boss, as indicated by its asymmetrically raised external surface and featureless, gently concave visceral side (Fig 4G–L). As preserved, the element is pentagonal but three of its edges are broken, one is not prepared, and only one is natural, presenting a complex sutural surface (Fig 4H). The suture has an irregular structure and it faces slightly towards the exterior of the shell (Fig 4G). It is ambiguous whether it represents a typical intercostal suture or a suture between the elements of the dermal carapacial mosaic observed in *Chinlechelys tenertesta* and *Proterochersis* spp. [9,19,21]. The pleural boss takes the form of a subhorizontal, elongated shelf, which is typical for *Proterochersis* spp. and *Keuperotesta limendorsa* (Fig 2C, D; particularly well-expressed in *Proterochersis porebensis*) [16,18,19,22] and probably australochelyids (Fig 2G) [7] but differs from *Proganochelys quenstedtii*, which has either poorly-pronounced or rounded, tubercle-like pleural bosses (Fig 2E, F) [4,33,73,119,122].

PRC 197 is a triangular in cross-section fragment likely representing a carapace periphery (Fig 3J′, K′). It preserves no sulci so its precise position in the shell is unknown. TF 1440-5a, described and figured by Broin *et al.* [1] could not be located for this revision and is apparently lost. However, published photographs and drawings ([1]: figs 1A, 2A) and surviving cast (MNHN, Fig 5A–C) reveal that the layout of sulci corresponds well with the anterior part of the proterochersid carapace (compare, e.g., *Proterochersis porebensis* ZPAL V. 39/49, Fig 5D [16,18]): the lateral right edge of the first vertebral scute, the anteromedial edge of the first right pleural scute, and parts of the second and third right marginal scute, rather than the posterolateral part, as suggested by Broin *et al.* [1]. The medial edge of the preserved presumed first vertebral scute area bears a sulcus-like groove but it is not clear from the cast whether this represents a genuine scute sulcus (in such a case the specimen should be interpreted differently), a break, or a sutural edge with more medially positioned elements (e.g., parts of the carapacial dermal mosaic [19,21]). The visceral surface of the specimen, which bears an irregular embayment roofed by a much thinner bone than the surrounding regions and interpreted by Broin *et al.* [1] as a set of sockets receiving distal ends of ribs, is more troublesome in interpretation. However, the carapace of *Proterochersis* spp. is at least in some places two-layered thanks to the contribution of a mosaic of supernumerary dermal ossification, and the proximity of the nuchal region is one of such places – this region was shown to form in *Proterochersis porebensis* a dorsally-roofed embayment receiving the first costal [19,21]. It seems, therefore, possible that the irregular pits or sockets in the visceral surface of TF 1440-5a might have received the proximal rather than distal parts of anterior costals and, perhaps, the anterior end of the hyoplastral (axillary) buttress. This is supported by their diverging rather than converging or parallel layout (Fig 5C) and the different organization of the costals and dermal bones (peripherals, carapacial dermal mosaic elements) in the bridge and posterior parts of the carapace in *Proterochersis* spp. [19]. The position of the most prominent embayment, the bottom one in Broin *et al.* [1] figures, relative to the sulci pattern agrees with the position of that buttress in *Proterochersis* spp. and the presence of a ventral/visceral bulging at the base of marginals (the surface of which is damaged) are, likewise, consistent. It must

be, moreover, noted that even if the embayments accommodated the distal ends of ribs, as interpreted by Broin *et al.* [1], there is no evidence that all three of them were located within a single peripheral (a feature considered unique by those authors) – similarly to most other Triassic turtles [4,6,7,16,21], the shell of *Thaichelys ruchae* tends to ankylose, making observation of most bone borders impossible. There is no apparent evidence of pits for the dorsal processes of scapulae, but in *Proterochersis* spp. these pits are located more medially than in *Proganochelys quenstedtii*, so they could have been located medially relative to the preserved part. Apparently, there was no evidence of supramarginals, rejecting the close affinity of the individual to *Proganochelys quenstedtii*. While the morphology of TF 1440-5a is not exactly replicated in any of the currently known specimens of *Proterochersis* spp., it is, therefore, generally explainable based on the grounds of the anatomy observed in the proterochersids, and differences may be an expression of taxonomic differences within the family.

**Plastron.** The material of *Thaichelys ruchae* includes five parts of the anterior plastral lobe: SM2015-1-001 (holotype, left, Figs 6, 8A, 9), SM2017-1-129 (left, Figs 7A–F, 8B, 9), MNHN.F.THA17 (left, Figs 7G–L, 8C), MNHN.F.THA21 (right, Figs 7M–R, 8D), and MNHN.F.THA18 (right, Fig 7S–X). SM2015-1-001 and SM2017-1-129 are comparable in size, MNHN.F.THA17 is intermediate in size, and MNHN.F.THA21 and MNHN.F.THA18 are smaller. The specimens show some differences in proportion and shape, and especially MNHN.F.THA18 has some juvenile features. Nevertheless, we agree with Broin [2] that SM2015-1-001 and SM2017-1-129 likely represent the same species, and refer MNHN.F.THA17, MNHN.F.THA21, and MNHN.F.THA18 to the same taxon, especially that the gular region exhibits a considerable variability in those respects in Triassic turtles (Figs 8, 9) [4,18]. All five specimens preserve much of the gular (rounded, incomplete apically in SM2017-1-129 and medially in MNHN.F.THA21; the completeness of the medial part in other specimens is difficult to confirm). All specimens except MNHN.F.THA18 retain a complete extragular projection (spike-like, with minor damage to the tip in SM2015-1-001 and SM2017-1-129), as well as a smaller (SM2017-1-129, MNHN.F.THA17, MNHN.F.THA21) or larger (SM2015-1-001) part of the humeral scute area. MNHN.F.THA18 is broken at the base of the extragular projection and before the edge of the humeral scute. The ventral (external) surface of SM2015-1-001 (Fig 6B) is crossed close to its midlength by a nearly transverse (gently anterolaterally directed) groove possibly representing the humeropectoral sulcus, but this groove coincides with a transverse break, therefore it is not clear whether it is a genuine imprint of scute borders or an artifact.

SM2015-1-001 (Fig 6) is broken in several places and while some of these breaks may follow suture lines, such a correspondence is not obvious. One of the breaks of SM2015-1-001 runs obliquely, anterolaterally, across the area of the extragular scute (Fig 6A, B), which follows the natural sutural posterolateral edge of the epiplastron of *Proterochersis porebensis* ZPAL V. 39/404 [19,28]. This may hint at the presence of an additional ossification supporting the tip of the extragular projection in *Thaichelys ruchae*, as in *Proterochersis porebensis* [19,28], but in this case the evidence is conjectural and it cannot be unambiguously confirmed whether in *Thaichelys ruchae* such an ossification was present (but fused to the epiplastron of SM2017-1-129), variably present (absent in SM2017-1-129), or absent altogether (the break in SM2015-1-001 would then be an artifact). SM2017-1-129 (Fig 7A–F) appears to be disarticulated from the rest of the plastron roughly along the suture with the ento- and hyoplastron – the posteromedial edge seems to show some lamellar structure (Fig 7D, F), however, it is difficult to unambiguously confirm how closely it follows the natural border of the bone due to the white paint with the specimen number located in that place and residues of rock matrix. Locally spongiosa is exposed, indicating some damage. MNHN.F.THA18 preserves a ventromedially directed facet in roughly the same area of the gular scute (Fig 7T, X) but, likewise,

no clear sutural features are apparent. Broken, worn surfaces of the Huai Hin Lat Formation fossils are in some cases difficult to distinguish from natural borders, especially when covered by thicker layers of glue. Note that Broin's [2] assumption that *Proterochersis robusta* had no gular and extragular projections, but rather a rounded-off, smooth edge of the anterior plastral lobe, and that the epiplastra in Triassic turtles were weakly attached to the hyoplastra was subsequently proven incorrect (Fig 8G) [4,16,18,19,22,28]. It was probably based on earlier, generalized and partly fictious reconstructions [51] and examination of incomplete (SMNS 16442) and immature (SMNS 16603) individuals.

MNHN.F.THA17 (Figs 7G–L, 8C), MNHN.F.THA12 (Figs 7M–R, 8D), and MNHN.F.THA18 (Fig 7S–X) are overall consistent morphologically with SM2015-1-001 and SM2017-1-129 but capture additional aspects of variability. MNHN.F.THA17 preserves the most complete gular and extragular projections. In MNHN.F.THA21, the gular projection is damaged medially. MNHN.F.THA17 and MNHN.F.THA18 present the most complete dorsal epiplastral processes, although in both specimens the processes have broken off dorsal tips and in MNHN.F.THA17 the process is unnaturally deflected laterally, likely because of compression. MNHN.F.THA18 has poorly preserved ventral surface and broken off extragular projection, but its gular projection showcases a seemingly juvenile morphology, reminiscent of *Proterochesis porebensis* ZPAL V. 39/501 but more developed than in *Proganochelys porebensis* ZPAL V. 39/34 and *Proterochersis robusta* SMNS 16603 [2,16,18,28].

Neither of the anterior plastral specimens preserves the midline so there is no evidence for or against the presence of an intergular scute and it is difficult to convincingly establish how wide was the anterior plastral lobe and at what angle the preserved parts were situated relative to its long axis. As preserved, the gular scute of SM2015-1-001 appears to have been slightly narrower than the extragular projection (Figs 6B, N, 8A, D) whereas in SM2017-1-129 and MNHN.F.THA17 both structures are of nearly equal width (Figs 7B, H, 8B, C). Although the extragular is usually wider than the gular in *Proterochersis* spp. (Fig 8E–I) and *Proganochelys quenstedtii* (Fig 8K–N), at least in some specimens, as well as in *Odontochelys semitestacea* (Fig 8P), their width usually did not differ greatly, regardless of individual [2,4,14,16,18,123], so this difference is easily explained by the intraspecific variability. If the size of those scutes in proportion to the whole plastron size is assumed to be similar as in *Proterochersis* spp., and their layout is assumed to be roughly transverse (i.e., with the extragular located laterally rather than posteriorly to the gular), then the anterior lobe of the plastron would need to be similar in proportions as in *Odontochelys semitestacea* and *Proganochelys quenstedtii*, that is relatively narrow anteriorly [2,4,14,73]. However, a more proterochersid-like, wider anterior lobe can be achieved by a slight increase of the proportional size of the gular and extragular and a change of the angle, allowing a more posterior position of the extragular. Both scenarios appear plausible, but neither can be designated as preferred. The relative position and angle of the gular and extragular projections and scutes are variable in *Proterochersis* spp. (Figs 8E–I, 9) [18]. In the hypodigm of *Proterochersis porebensis* there is at least one specimen (ZPAL V.39/420, Fig 8F, 9) which could produce a similar morphology of the extragular being more posterior than lateral relative to the gular, but it is also disarticulated, so its exact *in vivo* configuration within the plastron is ambiguous and such a morphology is not common in *Proterochersis* spp. Notably, however, the extragular in ZPAL V.39/420 is pointy, similar as in *Thaichelys ruchae*. A small intergular was present in *Proganochelys quenstedtii* (Figs 8K, L, 9; "median scute" of Gaffney [4,73]). In the known australochelyids, the configuration the anterior plastral scutes is poorly visible, but there seem to be wide, paired elements laterally and a single, convex and apparently unpaired, mesially (Figs 8O, 9) [5–7]. The former, based on their morphology and location, most likely correspond to the extragulars. The latter may either represent fused gulars or an intergular scute. The presence of an unpaired intergular is

here tentatively favored due to the known occurrence of that scute in *Proganochelys quenstedtii* and lack of evidence of gular scute fusion in Triassic turtles, but better preserved material is necessary to confirm this hypothesis. The presence of the gulars in the Australochelyidae is for now left ambiguous – they could either be lost, present but small and devoid of distinct projections, or excluded from the edge of the anterior plastral lobe.

Regardless of a precise angle, in any plausible scenario, the gular projections of *Thaichelys ruchae* SM2015-1-001 and SM2017-1-129 had to be directed anterolaterally and the extragular projections – predominantly laterally, with their anterior edges straight or gently convex and likely slanted slightly posterolaterally (Figs 6, 7, 8A–D, 9). The observed variability in that regard is relatively minor, with MNHN.F.THA17 having the extragular projection directed very slightly anterolaterally (Figs 7H, 8E). The direction of the extragular projections was indicated by Broin [2] as separating *Thaichelys ruchae* (her *Proganochelys ruchae*) from *Proganochelys quenstedtii*, the latter considered to have its extragular projections directed anteriorly or anterolaterally and curved. While it is true that *Proganochelys quenstedtii* SMNS 16980 (Figs 8K, 9, 10F) [2,4,73] and *Odontochelys semitestacea* IVPP V 13240 (Fig 8N) [14,123] have horn-like, curved extragular projections with anteriorly directed tips, the morphology of those structures is variable. The extragular projections of *Odontochelys semitestacea* IVPP V 15639 [14] and *Proganochelys quenstedtii* SMNS 17204 (Figs 8L, 9, 10E) [4,73] are directed anterolaterally but not curved (their anterior edges are straight rather than anteriorly concave). In *Proganochelys quenstedtii* SMNS 17203 (Figs 8M, 9) [4] and SMNS 15759 (Figs 8N, 9) (broken-off part of the right epiplastron, but inclination of the extragular projection can be estimated based on the gularoextragular sulcus) they are predominantly laterally directed. In *Proterochersis* spp., the extragular projections are variable (Fig 8E–I, 9, 10C, D) [18]. They have nearly straight, in some cases gently convex or gently concave anterior edges and are usually less spiky than in *Odontochelys semitestacea*, *Proganochelys quenstedtii*, and *Thaichelys ruchae*, in some cases rounded-off, making their inclination less obvious. If the axis running from the meeting point of the gular, extragular, and humeral scutes to the most protruding part of the free edge is considered, they are predominantly anterolaterally or laterally directed. *Keuperotesta limendorsa* SMNS 17757 has only a part of the extragular preserved (Fig 8J) but its morphology is similar as in *Proterochersis* spp [28]. In the australochelyids, the extragular points are directed anterolaterally (Figs 8O, 9) [6,7].

Ventrally, the gular projection is gently convex in SM2015-1-001 (Figs 6, 8A, 9) and MNHN.F.THA21 (Figs 7M, 8D), gently concave in MNHN.F.THA17 (Figs 7H, 8C) and MNHN.F.THA18 (Fig 7T), and flat in SM2017-1-129 (at least its preserved basal part; Figs 7B, 8B, 9). Ventral flatness or gentle convexity of the gulars is typical for *Proganochelys quenstedtii* (Figs 8K–N, 9, 10E, F) and *Proterochersis robusta* (Figs 8I, 10C) [2,4,14,16,18,19] but in non-juvenile *Proterochersis porebensis* the anterior/anteromedial edges of the gular projections are deflected downwards, producing together a distinct, V-shaped (anteriorly concave) lip, usually followed posterolaterally by a gentle concavity (Figs 8E–H, 9, 10D) [4,14,16,18,19]. The latter morphology is similar to that observed in MNHN.F.THA18, although this specimen seems to have its gular projection relatively undeveloped, possible due to its young ontogenetic age. Conversely, the ventral surface of the extragular projection is flat in SM2017-1-129 (Figs 7B, 8B, 9) but gently concave in SM2015-1-001 (Figs 6, 8A, 9), MNHN.F.THA17 (Figs 7H, 8C), and MNHN.F.THA21 (Figs 7M, 8D) due to slight ventral deflection of its anterior edge, resulting in a slightly C-shaped cross-section (Figs 6F, 9). Such a ventral deflection of the anterior edge of the extragular process is common (albeit variable) in *Proterochersis porebensis* (Figs 8E–H, 9, 10D) and the australochelyids (Figs 8O, 9) but the extragulars were ventrally flat or gently convex in *Proterochersis robusta* (Figs 8I, 10C), *Keuperotesta limendorsa* (at least in the only preserved specimen; Fig 8J), *Proganochelys quenstedtii* (Figs

8K–N, 9, 10E–G), and *Odontochelys semitestacea* (Fig 8P) [2,4,6,7,14,16,18,19]. In contrast to *Odontochelys semitestacea* and *Proganochelys quenstedtii*, but similar as in *Proterochersis* spp. [2,4,14,16,18,19,22,73,123], the gularohumeral sulcus in *Thaichelys ruchae* is nearly transverse rather than distinctly V-shaped, and the posteromedial edges of the gulars did not extend far posterior relative to the extragulars.

The dorsal surface of the extragular projection is relatively uniform, convex, and smooth in SM2015-1-001 (Figs 5A, F, 9), SM2017-1-129 (Figs 7A, E, 9), MNHN.F.THA17, and MNHN.F.THA21 (Fig 7M). In those respects, accounting for the differences in the geometry of that element, it is similar as in *Proterochersis* spp. (Fig 9) and *Keuperotesta limendorsa* [16,18,19,28]. In *Odontochelys semitestacea*, *Proganochelys quenstedtii*, and the australochelyids, the apex of each of the extragular projections bears along its dorsal surfaces a distinct ridge towards the base of the respective dorsal epiplastral process (Fig 9) [2,4,6,7]. Such a ridge is absent in *Thaichelys ruchae*. The posterolateral portion of the dorsal surface of the extragular projection in *Proganochelys quenstedtii* is, moreover, distinctly concave (Fig 9) [2,4,118]. The degree of that concavity is varied. In SMNS 16980 the surface is large, subvertical, and saddle-shaped due to the anterior deflection of the extragular projections, taking in dorsal view a pinna-like form [2,4,118]. In SMNS 17204 it is only exposed on the left side and damaged, but its extent was probably similar as in SMNS 16980. In SMNS 17203 [4] and SMNS 15759 it is more horizontal, probably due to the more lateral inclination of the extragular projection in those specimens, but still distinct. In most specimens it can be mistaken as an anterolateral expansion of the body wall onto the extragular projection, but it is separated from the visceral surfaces by a clear ridge in SMNS 16980 and it is clearly rugose in SMNS 15759 (Fig 9). There is no comparable concavity in SM2015-1-001, whereas in SM2017-1-129, MNHN.F.THA17, and MNHN.F.THA21 the dorsoposterolateral region of the extragular projection is gently depressed but the concavity is much less distinct than in *Proganochelys quenstedtii* (Figs 7A, D, E, J, K, M, 9). In *Proterochersis* spp., *Keuperotesta limendorsa*, and the australochelyids, the dorsal surfaces of the extragular projections are completely convex until they reach the body wall boundary (Fig 9) [6,7,16,19,28].

SM2015-1-001 preserves only a small lateral fragment of the base of the dorsal epiplastral process (Figs 6A, C–G, 9), but the base is nearly complete in SM2017-1-129 (Figs 7A, C–F, 9), MNHN.F.THA17 (Fig 7G. I–L), MNHN.F.THA21 (Fig 7M, O–R), and particularly in MNHN.F.THA18 (Fig 7S, U–X). In all of those specimens it is evident that the process was located on an elevation, some distance away from the proximal (basal) edges of the gular and extragular scutes and separated from their anterodorsal surfaces of by a distinct flexure. This is clearly different from the morphology observed in *Proganochelys quenstedtii* and the australochelyids, in which the dorsal epiplastral processes seem to abut the body wall and the anterior faces of their bases lie in virtually the same plane as the anterodorsal surfaces of the gulars and/or extragulars, as if they constituted their continuation (Fig 9) [4,6,7]. The close association of the bases of the dorsal epiplastral processes and the body surface in *Proganochelys quenstedtii* seems to be particularly well evidenced by SMNS 15759, in which the lower part of the dorsal epiplastral process is rugose and has the same surface characteristics as the gular and extragular projections, despite being separated from them by a V-shaped sulcus, suggesting that it was covered by a distinct scute (Fig 9). On the contrary, in *Odontochelys semitestacea* and *Proterochersis* spp. the dorsal processes of the epiplastra seem to be located deeper, as they are separated from the gulars and extragulars by a smoother area of bone and/or a distinct (V- or S-shaped in the mediolateral aspect) flexure [14,16,18,19,28], as in *Thaichelys ruchae*. In SM2017-1-129 this separation is especially clear medially, where the body wall attachment is expressed as a triangular area bearing several low grooves and ridges (Figs 7A, C, 9), and in MNHN.F.THA21 (Fig 7S, U), in which the homologous area is completely

smooth and plainly distinct from the rugose scute-covered surface. The preserved portions of the dorsal epiplastral processes have oval cross-sections and gradually decrease in diameter dorsally, but neither of them preserves a natural dorsal tip.

Broin [2] considered the dorsal epiplastral processes in SM2015-1-001 and SM2017-1-129 to be widely-separated and their lateromedial position to be more reminiscent of *Proganochelys quenstedtii* than *Proterochersis robusta*. We disagree with that opinion. In undeformed specimens of *Thaichelys ruchae*, the dorsal epiplastral processes are located completely or nearly completely within the lateromedial extent of the gular scutes (Figs 6A, C, 7A, C, M, O, S, U, 9), therefore nearly certainly closer to the midline than to the lateral limit of the anterior plastral lobe, as in *Proterochersis* spp. (Fig 9) [16,19,28]. In *Proganochelys quenstedtii*, they are located at the level of the gularoextragular sulcus and typically about halfway between the midline and the lateral extent of the extragular scute (except SMNS 15759 and perhaps SMNS 17203 due to the length and lateral rather than anterolateral inclination of their extragular projections, although in both specimens the dorsal epiplastral processes likely occupied a more lateral position than in *Thaichelys ruchae*). In the australochelyids, the position of the dorsal epiplastral processes is similar as in *Proganochelys quenstedtii*, while in *Odontochelys semitestacea* they are located even more laterally [6,7,14]. Admittedly, the dorsal epiplastral processes in *Proterochersis* spp. are somewhat smaller than in SM2017-1-129 (Fig 9). Moreover, *Proterochersis* spp. and *Keuperotesta limendorsa* differ from *Thaichelys ruchae*, *Proganochelys quenstedtii*, and the australochelyids in the presence of distinct pits or excavations below the posterolateral parts of the dorsal epiplastral processes – these pits are partially (anteromedially) roofed by the bases of the processes and probably accommodated the acromions or served as attachment points for the acromial ligaments (Fig 9) [16,19,28]. This posterolateral excavation, together with more medial placement of the dorsal epiplastral processes is associated in *Proterochersis* spp. with formation of a distinct, virtually vertical, lip-like elevation of the scute-covered parts of the extragular relative to the more posterior part of the plastron [16,19,28], in contrast to the more rounded and smoother posterior limit of the dorsal epiplastral process-bearing elevation of *Thaichelys ruchae*. In *Keuperotesta limendorsa* SMNS 17757 the dorsal part of the preserved extragular is damaged, obscuring the original morphology. In *Thaichelys ruchae* (Figs 6A, D–F, 7A, C–G, I–M, O–S, U–X, 9) and *Proganochelys quenstedtii* (Fig 9) the bases of the dorsal processes are convex posterolaterally [2,4]. In *Palaeochersis talampayensis* there is some gentle concavity in that area and a low, but clear Y-shaped midline ridge connecting the dorsal epiplastral processes and the posterior entoplastral process, but there are no distinct pits and there is little to no dorsal roofing above the concavities (Fig 9) [6]. *Waluchelys cavitesta* seems to exhibit a similar morphology [7]. *Proganochelys quenstedtii* and the australochelyids differ from *Thaichelys ruchae* and *Proterochersis* spp. also in a distinct mediolateral flaring of the bases of their dorsal epiplastral processes, resulting in their nearly triangular shape in the anteroposterior aspect and apparent continuity with the dorsolateral edges of the extragular projections (Fig 9) [4,6,7]. In *Proganochelys quenstedtii* there is some variability in the degree of that flaring – it is very clear in SMNS 16980 and least pronounced in SMNS 15759, but the transition between the lateral edge of the dorsal process and the dorsolateral edge of the extragular in general is relatively smooth. In contrast, in *Thaichelys ruchae* (Figs 6C–E, 7C, D, O, P, U, V, 9) and, particularly, *Proterochersis* spp. (Fig 9), the contour of those structures in the anteroposterior aspect is broken by a distinct angle and the dorsal epiplastral processes are clearly separated laterally from the tip of the extragular projections [28].

The posterior part of the entoplastron, including the posterior entoplastral process, is not preserved in any specimen of *Thaichelys ruchae*. The visceral surface of the hyoplastron, as preserved, is rather featureless and gently concave.

MNHN.F.THA12 (Fig 11A–F) is identified here as a partial hyoplastron. It is a triangular fragment of bone, thinner medially than laterally, with a ridge on the external surface at the meeting point between the main plastral plate and the bridge. A ridge-like posterior continuation of the anterior plastral lobe in the anterior part of the bridge is the most distinct in *Odontochelys semitestacea*, *Proterochersis* spp., and *Keuperotesta limendorsa*, in which the ventral surface of the main plate of the hyoplastron approaches the bridge at a nearly right angle, but it is less marked in *Proganochelys quenstedtii* (Fig 10C–G) [4,16,18,19,22,119,124]. In *Thaichelys ruchae* it seems to have been intermediate in extent, with both surfaces meeting at an obtuse angle but producing a distinct ridge.

The contact between the plastron and peripherals is broken and partly obscured in SM2017-1-125 by misplaced bone fragments, sediment residues, and glue (Figs 1F–J, 10A). We could not confirm the presence of a ligamentous connection between those structures, proposed by Broin [2] – the bridge could be fully ossified, just broken. Likewise, the presence or absence of inframarginals cannot be categorically confirmed. Considering the distance between the marginal scute edges and the lateral edge of the femoral scute, the bridge (and, thus, the inguinal notch) appears comparably narrower mediolaterally than in most Triassic turtles, but the difference is not striking (Fig 10) [4–7,16,18,19,22,51,73,119,120,122,124,125–127]. Its posterior edge is inclined more posterolaterally than laterally (transversely), more in line with *Proganochelys quenstedtii* (Fig 10E–G) [4,73,119,120,122,124,125–127] and the australochelyids (Fig 10H) [5–7,124] than *Proterochersis* spp. (Fig 10B–D) or *Keuperotesta limendorsa* [16,18,19,22,51,124]. However, the angle between the posterior edge of the bridge and the lateral edge of the posterior plastral lobe (partially obscured by a rock matrix residue) is acute (approaching right) and well-marked rather than obtuse and smooth (Figs 1F, 10A), which is observed in *Proterochersis* spp. (Fig 10B–D) and *Keuperotesta limendorsa* but not in other Triassic turtle taxa (Fig 10E–H) [4–7,16,18,19,22,28,51,73,119,120,122,124,125–127]. Moreover, the posterior edge of the bridge is deflected posterodorsally, forming a moderately developed inguinal buttress, comparable as in *Proterochersis* spp. and *Keuperotesta limendorsa* [16,19,28], more prominent than in *Proganochelys quenstedtii* [4,73,119,122,126,127], but less prominent than in the australochelyids [6,7]. Unlike in *Proganochelys quenstedtii* (Fig 10E–G), there is also no substantial posterolateral extension of the inguinal buttress [4,73,119,120,122,124,125–127] – the inguinal notch is less than one marginal deep anteroposteriorly, similarly as in *Proterochersis* spp. (Fig 10B–D) and *Keuperotesta limendorsa*, and possibly (at least when it comes to its ventral face) australochelyids (Fig 10H) [5–7,16,18,19,22,51,124].

The plastral scute sulci in the bridge region of SM2017-1-125 are deep, wide, and well-marked, and the scutes had a gently convex rather than depressed surface (Figs 1F, 10A). This is consistent with the morphology observed in most specimens *Proterochersis robusta* (Fig 10B, C) and some specimens of *Proterochersis porebensis* (others having flatter scute surfaces; Fig 10D) [16,18,19,22,51,124] but not *Keuperotesta limendorsa* (with flat plastral scute surfaces) [16,124], *Proganochelys quenstedtii* (with flat or gently depressed plastral scute surfaces and usually much shallower and poorly-defined plastral scute sulci, Fig 10E, F; except MB.1910.45.2, Fig 10G, and the lost plastron of SMNS 15759 with better defined but narrow sulci) [4,119,124], and the australochelyids (also with poor sulci definition and flat plastral scute surfaces; Fig 10H) [5–7,124]. Identifiable are at least three scutes: a small fragment of the femoral, at least one abdominal, and one or two preceding scutes – the preservation of the anterior portion of the bridge is poor and it is ambiguous whether the small, slightly displaced fragment observed in that region was broken off along the scute sulcus, which could constitute a weak point, or was covered by the same scute as the neighboring, more posterior part of the bone. However, the proportions of the preserved part fit the double abdominal configuration of *Proterochersis* spp. and *Keuperotesta limendorsa* (Fig 10A–D) rather than the single

abdominal of *Proganochelys quenstedtii* (Fig 10E–G). Most notably, the presumed second abdominal is very short anteroposteriorly, its length does not exceed the length of a single marginal scute and is relatively uniform throughout the bridge width, in contrast to much broader and medially tapering abdominal of *Proganochelys quenstedtii* but in accord with the anatomy of *Proterochersis* spp. and *Keuperotesta limendorsa* [4,16,18,19,22,51,73,119,124]. The preceding scute, that is the presumed first abdominal, if its anterior limit is accepted to coincide with the observed break, would be very slightly shorter. This also fits the proportions and scute connection pattern in *Proterochersis* spp., *Keuperotesta limendorsa*, and *Odontochelys semitestacea*, also having two pairs of abdominal scutes of subequal length [14,16,18,19,22,51,123,124]. In such a case, the more anterior, broken off fragment would represent the posterolateral portion of the pectoral scute area. Otherwise, if only one abdominal scute is accepted, the extent of the pectoral would reach unusually far posteriorly and cover about two thirds of the bridge length – in *Proganochelys quenstedtii* the pectoroabdominal sulcus is located just anteriorly to the midlength of the bridge (Fig 10E–G) [4,73,119]. Plastral sulci in the known australochelyid specimens are too indistinct to allow determination of the number and position of the abdominals (Fig 10H) [5,6,124,128].

The bone in the preserved area of the femoral scute of SM2017-1-125 is thickened compared to the rest of the preserved plastron (Fig 1J). A bulbous thickening in that area is particularly distinct in *Proterochersis* spp. and *Keuperotesta limendorsa*, but the edge of the anterior part of the posterior plastral lobe is, to a lesser extent, also thickened in other Triassic turtles [4,6,7,16,19,28,51].

Unlike Broin [2], we refrain from equating the breaks in the plastron of SM2017-1-125 with suture lines, noting that shells of Triassic turtles are typically ankylosed [4,6,7,16,21]. Therefore, the number of mesoplastra in *Thaichelys ruchae* remains ambiguous. However, the number of mesoplastra and abdominal scutes in Triassic pantestudinates appear correlated [4,14,19,51,123], therefore two pairs of mesoplastral may be inferred if two pairs of abdominals are accepted.

MNHN.F.THA20 (Fig. 11G–L) is a plate-like, polygonal element with at least one (presumably lateral) natural edge, which we interpret as the left xiphiplastron. This identification is based on its thickened lateral part, consistent with the morphology of the posterior plastral lobe in Triassic turtles [4,6,7,16,19,28,51], including SM2017-1-125, its concave external surface (rather than convex, as in the anterior plastral lobe), and the presence of a diagonal, posterolaterally inclined groove (barely detectable due to rock residue), which we interpret as the femoroanal sulcus. Although most edges of the bone are damaged, its general outline seems to be consistent with the shape of the xiphiplastron in *Proterochersis* spp. (e.g., *Proterochersis robusta* SMNS 16442, *Proterochersis porebensis* ZPAL V. 39/170) [19]. Unfortunately, the visceral surface of the specimen is covered in what appears to be a thick layer of glue and rock residue, disallowing determination of whether the bone formed a proterochersid-like sutural connection with the pelvis [16,19,20,23,28,51,121,129,130]. Posteriorly, the specimen displays a posteromedially directed notch-like facet, which roughly corresponds to the position and form of the suture with the supernumerary (caudal and intercaudal) plastral ossifications of *Proterochersis* spp. [19], but the preservation is not good enough to unambiguously establish if this is indeed a worn sutural surface or a break.

**Humerus.** MNHN.F.THA16 (Fig. 12A–F) is a complete but compacted in the capitular-intertubercular plane left humerus. The bone is 54 mm long and presents juvenile features, but due to the lack of a larger sample it is uncertain whether this is indeed an effect of a young ontogenetic age of the individual or neotenic-like adult morphology of the species.

The proximal expansion is about 1.5 the width of the distal expansion and has an ovoid outline in the capitular and intertubercular view (Fig 12A, B). The proximal edge is uniformly convex, with no notches separating the humeral head and the lateral and medial processes. A similar uniformity of that edge is present in *Pappochelys rosinae*, *Eorhynchochelys sinensis*, juvenile *Odontochelys semitestacea*, and juvenile *Proterochersis porebensis* [11,12,14,15,28,54,131]. The humeral head is oval (wider than long) in capitular view and slanted radiodistally, similar as in *Proterochersis porebensis* and *Palaeochersis talampayensis*, so its ulnar portion extends the furthest proximally [6,19,54,131,132]. In undistorted specimens of *Proganochelys quenstedtii*, the humeral head is more symmetrical and its proximal-most point is located near the middle of its width [4,33,54,131–133]. There is no evidence of a shoulder, but it is uncertain whether this state is representative of the adult morphology, or a juvenile character – at least in *Proterochersis porebensis*, the presence of the shoulder is ontogenetically dependent [54,131,132]. Both the lateral and medial processes are proximally shorter than the humeral head and the medial process extends slightly further proximally than the lateral process (Fig 12A–D). Due to compaction, the intertubercular fossa is very indistinct and only the lateral process is marked intertubercularly as a proximoradially directed ridge (Fig 12B, E). Despite compaction, the lateral process seems to be connected to the humeral head by a relatively thick and proximally flattened bridge of bone, thicker than the connection between the humeral head and the medial process (Fig 12D, E). This morphology is shared with *Proterochersis porebensis*, whereas in *Proganochelys quenstedtii* the lateral process and the humeral head are connected by a thinner ridge of bone [4,28,54,132]. The radial edge of the lateral process bears a ridge-like, proximodistally extending, capitularily deflected deltopectoral crest (Fig 12 A, D, E). As in *Proterochersis porebensis*, the capitular surface of the base of the lateral process is shallowly concave forming a proximodistally elongated subcapitular fossa (Fig 12A, D) [28,54,132].

As preserved, the shaft is D-shaped in cross-section, with a flattened intertubercular surface and a convex capitular surface. It is distinctly constricted ulnoradially. Although it is broken near the midlenght and its exact shape is partially obscured by rock matrix remains and glue, the narrowest point of the shaft appears to be located around its middle (Fig 12 A, B). Due to compaction, the capitulo-intertubercular curvature of the shaft is only barely noticeable (Fig 12C, D).

The distal expansion is triangular in the capitulo-intertubercular aspect and bears capitularily and intertubercularily shallow, teardrop-shaped (narrower distally) fossae (Fig 12A, B). It is ulnoradially narrower than the proximal expansion and it uniformly broadens distally at a relatively low angle – in that regard it resembles *Palaeochersis talampayensis* and juvenile specimens of *Odontochelys semitestacea* and *Proterochersis porebensis* rather than *Pappochelys rosinae*, *Eorhynchochelys sinensis*, *Proganochelys quenstedtii*, and adults of *Odontochelys semitestacea* and *Proterochersis porebensis*, which show a more rapid distal broadening [4,6,11,12,14,15,28,33,54,131,132]. On the capitular surface of the distal expansion, close to its radial edge, there is an open ectepicondylar groove (Fig 1A, D, F). As preserved, the radial epicondyle projects radially a small but distinct, triangular in the capitulointertubercular aspect process, whereas the ulnar epicondyle is rounded off (Fig 12A, B). The intercondylar groove is barely marked (Fig 12A, F). The distal articulation surface is nearly uniform, its division into the capitellum and trochlea is poorly marked (Fig 12B, F).

**Radius.** MNHN.F.THA19 (Fig 12G, H) is the distal part of a radius. The shaft is relatively straight and rounded in cross-section. The bone differs from *Proganochelys quenstedtii* in the lack of a distinct longitudinal ridge along the medial edge of the dorsal surface of the shaft and proximal part of the distal end [4]. The distal end is flattened, triangular in the dorsoventral aspect, and deflected laterally, so its medial edge is gently convex and the lateral edge is gently

concave along the proximodistal axis. The body side it belongs to is difficult to establish, but based on the slight concavity of one surface (interpreted here as ventral), it is tentatively considered here to be the right radius. The presumed dorsal surface is flat, as in other Triassic testudinates, rather than convex, as in *Eorhynchochelys sinensis* and *Odontochelys semitestacea* [4,6,14,15,28]. There is no distinct process for the radioulnar ligament. The distal articular surface is weakly subdivided into two facets, for the first distal carpal and for the intermedium and medial centrale. The bone terminates distally in a short (possibly slightly damaged apically) styloid process.

**Ilium.** SM2017-1-130 (Figs 13, 14A, B) is a possible fragmentary left ilium tentatively referred to *Thaichelys ruchae*. As preserved, the specimen is 3 cm tall but is missing both the ventral and dorsal ends.

The ventral end of the specimen is expanded but broken off along a nearly flat plane above the acetabulum. A part of the base of the lappet projected by the ilium, which dorsally roofed the acetabulum, is preserved but the lappet itself is broken off (Fig 13A, B, D, F, 14A, B). Nonetheless, the acetabulum was apparently deflected slightly posteriorly relatively to the lateral surface of the bone, as in on the Triassic turtles [4,6,7,9,16,23,28,51,119].

The iliac neck is slender compared to that of *Odontochelys semitestacea*, *Proganochelys quenstedtii* (Fig 14G, H), and the australochelyids (Fig 14I, J) [4,6,7,14,119], but somewhat less so than in *Chinlechelys tenertesta*, *Keuperotesta limendorsa*, and *Proterochersis* spp. (Fig 14C–F) [9,16,23,28,51]. It is compressed mediolaterally, particularly in the posterior part (Fig 13D, E). Its lateral surface is convex (Fig 13A, 11A) and the medial surface increasingly concave towards the dorsal end (Fig 13C, E). Approximately halfway along the preserved height, the anterior part of the medial face produces a wide, vertical ridge (Fig 13C, E). The dorsal part of that ridge is damaged but it most likely continued towards the first sacral rib [16,23,28]. The rib, however, was apparently attached much above the acetabulum, as in *Keuperotesta limendorsa* and *Proterochersis* spp. (Fig 14F) but unlike in *Proganochelys quenstedtii* (Fig 14H), *Chinlechelys tenertesta*, and the australochelyids (Fig 14J) [4,6,7,9,16,23,28,51,119]. Unlike in the australochelyids (Fig 14H), the bone is not distinctly expanded anteromedially [6,7].

The dorsal portion of the bone is damaged and worn along the whole anterior, dorsal, and posterior edge and in the dorsolateral part, but it is clearly compressed mediolaterally (Fig 13D, E) and expanded slightly anteriorly and more strongly posteriorly (Figs 13A, C, E, 14A. Due to the damage, it is not evident whether it formed an osseous connection with the carapace. In its general form, however, it corresponds much better to the well-preserved ilia of the remaining proterochersids than other Triassic turtles (Fig 14) [4,6,7,16,23,28,51,119].

**Indeterminate elements.** SM2017-1-126 (Fig 15A–D) is a small bone fragment interpreted by Broin [2] as a lateral part of hypoplastron, with surfaces for ligamentous attachment to peripherals, but such an identification of the specimen is not certain. The fragment is gently curved, with the convex surface mostly featureless and the concave surface bearing two, nearly parallel ridges (Fig 15B–D). One of the edges is indeed sinuous and undulating, with the surface divided into smaller, concave fields (Fig 15D), but the state of preservation makes it difficult to evaluate how much of this morphology is natural and undamaged. Moreover, the ridged face is unlike the visceral surface of the bridge of Triassic turtles [4,6,7,16,19,28,119,120,122,125,126]. It is possible that the fragment represents a carapace piece with the visceral ridges being rib shafts and broken along a sinuous (e.g., vertebropleural) sulcus or contact with an element of the carapacial dermal mosaic [19,21], but the morphology is ambiguous. Likewise, SM2017-1-127, a small but relatively thick bone fragment without natural edges but with a wide, transverse groove (sulcus?) on one (possibly external) surface, and an undulating, narrower but deeper groove on the other (possibly visceral) surface (Fig 15E–H), considered by Broin [2] a possible costal, is here deemed

indeterminate. It may thus come from either the carapace or plastron. SM2017-1-133 (Fig 15O, P), SM2017-1-134 (Fig 15Q, R), and SM2017-1-135 (Fig 15I, J) also most likely represent unidentified shell fragments. SM2017-1-134 (Fig 15Q, R) is an elongated element which decreases in thickness along its long axis and bears a wide, longitudinal sulcus-like groove on one of its surfaces (Fig 15Q). These characteristics may suggest that it either represents a fragment of a carapace margin or a part of the plastron (in *Proterochersis* spp. the plastron becomes gradually thinner towards the midline in the central part).

PRC 199 (Fig 15W, X) is a possible fragment of the carapace consisting of parts of two adjacent costals joined along the intercostal suture. Remains of a partially fused, interdigitating suture are visible in that specimen externally (Fig 15W). Its visceral surface bears a groove (Fig 15X) comparable to the grooves for intercostal nerves or vasculature observed in other turtles and usually associated with the intercostal sutures but retained even after complete shell ankylosis [4,16,19,21].

MNHN.F.THA10 is a flat, thin, polygonal plate. It has worn, slightly undulating rims with a seemingly decreased vascularity, suggestive of the presence of sutural edges, and a somewhat radial pattern of vascular ornamentation on its external surface. These characteristics are consistent with the elements of the carapacial dermal mosaic of *Proterochersis* spp. and possibly *Chinlechelys tenertesta* [8–10,19,21]. Unfortunately, the specimen is incomplete and thus its identification in ambiguous.

PRC 201 is an approximately 2 cm long bone piece with a rounded edge, distinct taper, and striated surface (Fig 15A′, B′) resembling developing plastral bones of juvenile turtles [4,18]. The element is incomplete and preserves no sulci, so its identification is uncertain.

PRC 196 (Fig 15S, T), PRC 198 (Fig 15U, V), and PRC 200 (Fig 15Y, Z) are probable turtle shell fragments preserving no diagnostic details. The material from Huai Kee Tom includes also multiple other specimens which may constitute turtle shell fragments but the identification of which is ambiguous.

SM2017-1-131 (Fig 15K, L) is an about 2.5 cm long, nearly cylindrical (oval in the cross-section) bone piece with no natural ends and preserving little morphological information. It may represent a fragment of the dorsal epiplastral process, dorsal scapular process, or a shaft of a limb bone, or a portion of a non-turtle rib. Its referral to *Thaichelys ruchae* is tentative, although it has no visible medullary cavity, which diminishes the possibility that it is an appendicular bone of a nonpantestudinate tetrapod. SM2017-1-136 (Fig 15M, N) is another elongate bone fragment with an expanded and terminally concave end. Only one side of the specimen preserves natural surfaces (Fig 15M), making its identification difficult. It may represent a part of the shaft of a limb bone or a part of the dorsal epiplastral process, but its taxonomic identity as a turtle is not certain.

The original specimen TF 1440-5b [1] is lost but we were able to examine the cast (MNHN.F.THA22.2, Fig. 15E′–F′). The specimen is difficult to interpret. Broin et al. [1] considered it to preserve a short section of the natural edge, but at least in the cast this part (although indeed very thin) appears to be broken rather than separated along a suture or free. The resulting taper and the presumed ventral deflection of the thinnest part bring to mind anterior peripherals of proterochersids, but at least in the cast this side of the thin portion is smoother, bears only few enlarged vascular oppenings and grooves, and appears to represent a visceral surface rather than a scute-covered underside of a peripheral. The presumed ventral side at the opposite end of TF 1440-5b is swollen, rough, and bears marks of broken or eroded bone surface. The most distinctive feature of TF 1440-5b is the presence of a deep and broad sulcus on its external surface. This sulcus is off-center, roughly follows the long axis of the specimen, and covers about the third of its external surface's width at the edge opposite to its thinnest border. It is subtly bowed (convex towards the thinnest edge) but shows no distinct

undulation, and just next to one of the short edges, it deflects at an angle of about 125 degrees. The scute area just next to the sulcus is raised. It seems likely that the specimen represents a part of the carapace, possibly from the posteromedial region (junction between the third and fourth vertebral scutes and the fourth pleural scute), where the bones in *Proterochersis porebensis* can be very thin [16,19]. However, its precise location in the shell is uncertain.

**Ornamentation.** Broin [2] noted that the surface ornamentation in *Thaichelys ruchae* is different from *Proganochelys quenstedtii*. We agree with that notion – the external surfaces of all specimens from Thailand bear a variably distinct pattern of fine branching and anastomosing vasculature imprints, with some areas smoother than others, while adult specimens of *Proganochelys quenstedtii* tend to have a much coarser, larger in scale and deeper vascular groove network (Figs 2E, F, 8I, J, 9, 10E–G) [2,4,73,119]. Moreover, *Proganochelys quenstedtii* tends to have a more exaggerated undulation of carapace scute sulci (Fig 2E, F) [4,73,119]. On the contrary, Broin [2] described the shell surface of *Proterochersis robusta* as finer, more granular, and even slightly pitted. While the former two characteristics accurately describe most specimens (Figs 2B–D, 5D, 8C–G, 9, 10B–D), the subtle pitting observed in SMNS 16442, studied by Broin [2], is most likely a result of superficial damage rather than a natural feature [19]. Moreover, the surface ornamentation in *Proterochersis* spp. is highly variable – the shells can be smooth or rough and bear vascular imprints and/or miniscule, sparsely or densely packed, equidimensional or elongate tubercles [16,18–21]. This variability is evident not only between specimens, but sometimes even within a single specimen, and depends on the part of the shell, ontogenetic age, individual variation, and preservation. Even though an average specimen of *Thaichelys ruchae* has a somewhat smoother appearance than an average specimen of *Proterochersis* spp., the ornamentation types observed in the former generally overlap with the variability seen in the latter and their representation can be found in the hypodigm of *Proterochersis porebensis*. As such, the shell ornamentation is not particularly diagnostic for Triassic turtles but is not inconsistent with the proterochersid affinities of *Thaichelys ruchae*, especially when accounting for an influence of the locality- or formation-specific taphonomic and diagenetic factors on bone surface preservation. Bone surfaces are generally poorly preserved in the known australochelyid material [6,7] but at least in *Waluchelys cavistesta* the shell ornamentation appears to be less coarse than in *Proganochelys quenstedtii* [7]. *Chinlechelys tenertesta* has a smoother shell with numerous vascular openings and short, usually not branched and non-anastomosing vascular grooves [8–10].

## Phylogenetic analysis

The phylogenetic analysis resulted in 79 trees (best score 1054, consistency index = 0.328, retention index = 0.75; see Supporting Information for the complete topology and lists of synapomorphies). In all trees, *Thaichelys ruchae* was recovered in a clade with *Chinlechelys tenertesta*, *Keuperotesta limendorsa*, and *Proterochersis* spp., recognized here as the *Proterochersidae* (Figs 16, S1, S2). This clade is basal relative to, consecutively, *Proganochelys quenstedtii*, the Australochelyidae (polytomy of *Australochelys africanus* + *Palaeochersis talampayensis* + *Waluchelys cavitesta*), the clade of *Eileanchelys waldmani* (*Heckerochelys romani* (*Indochelys spatulata* (*Kayentachelys aprix* + *Condorchelys antiqua*), and the *Mesochelydia* Joyce, 2017 [34]. The strict consensus grouped all the proterochersids, *Thaichelys ruchae* included, in a polytomy and produced several more polytomies crownward to the *Eileanchelys waldmani* clade (Figs 16, S1). The majority rule (50%) tree improved the resolution significantly inside the *Mesochelydia* but only slightly stratified the *Proterochersidae* into *Thaichelys ruchae* + *Chinlechelys tenertesta* + (*Proterochersis* spp. + *Keuperotesta limendorsa*) and kept the three

australochelyids in a polytomy (Figs 16, S2). The resulting general topology is thus consistent with that recovered before by Szczygielski and Sulej [19] and one of the variants obtained by Sterli *et al.* [7] (the other one placing the proterocherids crownward relative to *Proganochelys quenstedtii*), but includes more Triassic taxa than any previous analysis. The support is, unfortunately, weak for most nodes – among early Mesozoic (pre-Late Jurassic) taxa, bootstrap values above 50 were only recovered for the Australochelyidae (78) and *Mesochelydia* (60) (Figs 16, S1).

In the strict consensus topology (Figs 16, S1), the *Proterochersidae* are supported by eight synapomorphies: Costal E (10 pairs of costals); Dorsal epiplastral process A (lateroventro-posterior excavation present); Dorsal epiplastral process D (located closer to the midline than to the lateral edge of the anterior plastral lobe, not extending significantly beyond the lateral extent of the gular scutes); Extragular process B (dorsal, anterolaterally or laterally directed ridge extending from the tip of the extragular process to the base of the dorsal epiplastral process absent or weak); Ilium A (elongated iliac neck present); Intercaudal and caudal scutes (present); Pelvis A (pelvis-shell attachment sutured); Xiphiplastron A (distinct anal notch present). In the same topology, the clade of *Proganochelys quenstedtii* and more derived turtles is supported by five synapomorphies: Abdominal B (one pair); Cervical vertebra L (eighth presacral vertebra has an intermediate (transitional) cervico-dorsal morphology, can be sutured to the carapace but not to the succeeding vertebrae); Coracoid (flat, rectangular or with a distinctly angular posteromedial edge); Femur A (articular surface of femoral head rectangular to oval in dorsal view; and Mesoplastron A (one pair of mesoplastra with medial contact). The more crownward clade of the Australochelyidae + *Mesochelydia* is supported by seven synapomorphies: Cervical vertebra L (eighth presacral vertebra free from the carapace and succeeding vertebrae, movable); Pectoral girdle A (horizontal plate with a dorsal process, not triradiate, bridge closing coracoid foramen narrower than the width of the coracoid foramen), Pterygoid A (pterygoid teeth absent); Pterygoid B (basipterygoid process present and sutured articulation); Quadrate A (flooring of the cranioquadrate space by the pterygoid, but the pterygoid does not cover the prootic); Quadrate B + C (development of the cavum tympani shallow, but anteroposteriorly developed); Vomer C (vomerine and palatine teeth absent). The Australochelyidae are supported by five synapomorphies: Extragular edge C (ventral deflection of the anterior edge present, the process concave ventrally and comma shaped in cross-section); Gular process (none, the anterolateral edge of the anterior plastral lobe even); Opisthotic B (depressions for musculature present); Parietal D (overhanging process of the skull roof present); and Peripheral bones (posterior peripheral bones with internal cavity). The *Mesochelydia* exclusive of Triassic taxa are supported by 19 characters: Coracoid (columnar, at least at its base); Dorsal epiplastral process D (located closer to the midline than to the lateral edge of the anterior plastral lobe, not extending significantly beyond the lateral extent of the gular scutes – acquired independently from the *Proterochersidae*); Entoplastron A (anterior entoplastral process absent); Extragular edge B (lateral or anterolateral tip rounded); Extragular process B (dorsal, anterolaterally or laterally directed ridge extending from the tip of the extragular process to the base of the dorsal epiplastral process absent or weak); Hypoischium A (absent); Ilium A (elongated iliac neck present – acquired independently from the *Proterochersidae*); Lacrimal A (absent); Opisthotic D2 (Processus interfenestralis present, small, reaching the floor of cavum acustico-jugulare); Pectoral girdle A (triradiate, bridge closing coracoid foramen); Pelvic girdle (ischium covered ventrally by the plastron); Plastron A (connection between carapace and plastron ligamentous); Plastron B (central plastral fontanelle present); Premaxilla A (external nares united); Pygal notch (absent); Quadrate B + C (deep and anteroposteriorly developed); Recessus scalae tympani A (well developed); Supramarginal A (absent); Vomer A (single).

In the majority rule (50%) topology (Figs 16, S2), the *Proterochersidae* as a whole are supported by three synapomorphies: Dorsal epiplastral process D (located closer to the midline than to the lateral edge of the anterior plastral lobe, not extending significantly beyond the lateral extent of the gular scutes); Extragular process B (dorsal, anterolaterally or laterally directed ridge extending from the tip of the extragular process to the base of the dorsal epiplastral process absent or weak); Ilium A (elongated iliac neck present). The clade of *Proterochersis* spp. + *Keuperotesta limendorsa* is further differentiated based on two synapomorphies: Dorsal epiplastral process A (lateroventroposterior excavation present) and Marginal B (serration of posterior marginal scutes in adults pronounced, rounded tips of underlying peripherals). The clades of *Proganochelys quenstedtii* and more derived turtles, the Australochelyidae, and the *Mesochelydia* are supported by the same synapomorphies as in the case of the strict consensus. The clade of the Australochelyidae + *Mesochelydia* is supported by an additional eighth synapomorphy: Marginal B (weak or no serration of posterior marginal scutes in adults).

Amniota indet: SM2017-1-132 is an asymmetrical, about 3 cm long, pipe-shaped element, consisting of a long, slender process and an expanded body with a flat, ovoid facet set at an angle to the long axis (Fig 17). The specimen is damaged and lacks unambiguous characters that would allow definitive identification. Therefore, although we provide two possible interpretations, both have to be treated as suggestions rather than proper referral. If the long axis of the specimen would be recognized as the craniocaudal one, then the form of the bone is most reminiscent of an atlas neural arch of kannemyeriiform dicynodonts (with the expanded body directed anteriorly) [134–136]. However, the overall shape is somewhat simpler. There is no distinct articulation facet for the proatlas and the flat facet, at least as preserved, shows no subdivision into the pre- and postzygapophysis – it is in its entirety directed anteroventromedially. Likewise, there is no preserved lateral process, but the area where its base would be expected is broken. The process resembles the dorsoposterior process of the kannemyeriiforms in the presence of three distinct faces comparable to the convex laterodorsal, subvertical medial, and concave ventral faces of the dicynodont dorsoposterior process.

Another possible interpretation, if the long axis of the specimen would be oriented transversely, is that the element represents a broken off fragment of a neural arch with the right prezygapophysis and damaged transverse process or rib fragment. The anterior inclination of the process makes it similar in shape to the proximal parts of dorsal ribs in some Triassic archosaurs, for example phytosaurs such as *Parasuchus* sp. [137] or aetosaurs such as *Stagonolepis* sp. [138], which would suggest one of the anterior dorsal vertebrae. At least phytosaurs have been already reported from the Huai Hin Lat Formation [43]. In any case, the specimen does not fit the morphology observed in pantestudinates and thus is treated here as an unidentified amniote.

## Discussion

Triassic pantestudinates present a significant amount of variability when it comes to the anatomy of their anterior plastral lobes, both inter- and intraspecifically. For that reason, this region of the shell may be useful for taxonomic identifications, at least at the genus level, but determination of the range of morphologies expected in a single species is difficult when the available material is limited. Fortunately, recent developments regarding the anatomy of that part of the shell [6,7,14,16,18,19,28,139] paint a much more detailed picture of its evolution and form, especially in comparison to the very limited data available at the end of the previous century [2,4,31,73]. The variability and partial morphological overlap between various Triassic turtle taxa allows, fortunately, an easier determination of homologies.

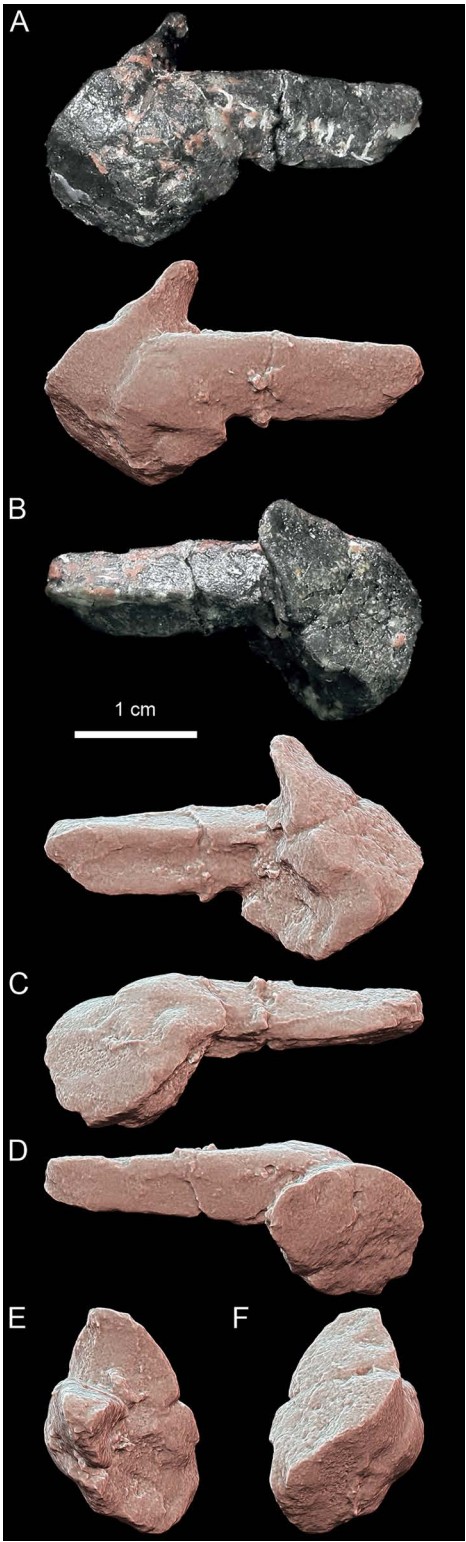

**Fig 17. Amniota indet. SM2017-1-132, probable fragment of a neural arch.** Precise orientation not determined.

In contrast to initial assessment, *Thaichelys ruchae* exhibits more characters typical of proterochersid turtles (Fig. 18) than of *Proganochelys quenstedtii*, but the combination of characters observed in that species is unique, especially when it comes to the anterior lobe of the plastron, and may be read as transitional between the less derived taxa such as *Odontochelys semitestacea* or the common ancestor of the *Proterochersidae* and *Proganochelys quenstedtii*, and the more derived and specialized *Proterochersis* spp. and *Keuperotesta limendorsa*. The relative reduction of the size and lateral restriction of the dorsal epiplastral processes in *Proterochersis* spp. compared to other Triassic turtles might have been thus achieved independently from the *Mesochelydia*, but the processes could have had a more plesiomorphic form in the less derived representatives of the proterochersid clade. *Thaichelys ruchae* is recovered in the results of our phylogenetic analysis as a member of the *Proterochersidae* in a polytomy with *Chinlechelys tenertesta* and outside of the clade of the European taxa (*Proterochersis* spp. + *Keuperotesta*). Although this result must be treated with caution due to the fragmentary nature of *Thaichelys ruchae* and, particularly, *Chinlechelys tenertesta*, it is consistent with the numerous characters observed in both the carapace and plastron. If correct, this basal position of *Thaichelys ruchae* with the *Proterochersidae* may mean that the presence of that taxon in Thailand could represent an initial wave of dispersal of late Triassic pantestudinates from eastern Pangea (modern-day eastern Asia; otherwise represented only by the non-testudinate pantestudinates *Eorhynchochelys sinensis* and *Odontochelys semitestacea* in the Carnian of China, with no Norian or Rhaetian taxa [14,15]) to central and western Pangea (modern-day Europe and North America), where they diversified and became much more common in the Norian [3,8–10,16–23,28,31–33,51,90,97,98,120,121,125–127,140,141], and eventually to southern Pangea (modern-day South America) in the late Norian and Rhaetian [5–7,139]. Note that in the Triassic the eastern part of Pangaea was partially separated from the rest of the continent by (Paleo-)Tethys and the land bridge connecting the two was located at high latitudes and mountainy, possibly hampering dispersal of continental tetrapods. Moreover, the reidentification of *Thaichelys ruchae* and *Chinlechelys tenertesta* ('*Proganochelys*' *tenertesta* of Joyce [34]; see also discussion in Szczygielski & Sulej [19] and Lichtig & Lucas [9]) as probable proterochersids removes the record of the genus *Proganochelys* from the modern-day Asia and the USA, making it a solely central Pangean (modern-day European) taxon (the proganochelyid turtle from the Norian of Greenland is treated here as a representative of the same clade but a generically separate taxon [31,32,141]). That means that the dispersal wave(s) from eastern Pangea must have included both the proterochersids, as an already distinct clade,

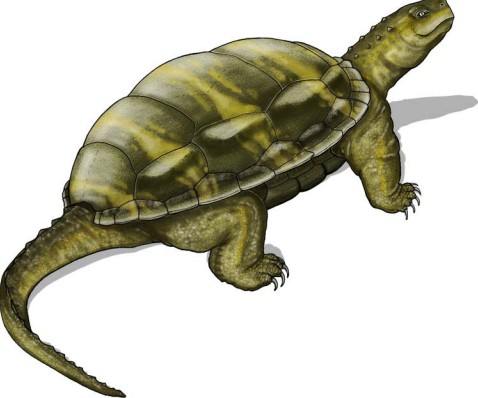

**Fig 18.** ***Thaichelys ruchae*, life restoration as a proterochersid turtle.** Digital drawing by Sita Manitkoon.

and the common ancestor of *Proganochelys quenstedtii* and the australochelyids (or early representatives of both lineages, if they diverged earlier). Probable less derived, non-testudinate stem turtles are known from the Permian South Africa [142–153] and the Middle Triassic of Germany [11–13,154] but in both cases they are separated by a significant temporal gap in the fossil record from the representatives of the radiation of true *Testudinata*.

The preserved material of *Thaichelys ruchae* suggests that this species could attain the somatic maturity at smaller body sizes than *Proterochersis porebensis*. Assuming proterochersid-like shell proportions, the size of SM2017-1-125 can be estimated to be only about 90% of *Proterochersis porebensis* ZPAL V. 39/34 (the smallest complete individual, carapace approximately 28 cm long), 50% of *Proterochersis porebensis* ZPAL V. 39/48 (holotype, carapace approximately 42.5 cm long), 42% of *Proterochersis porebensis* ZPAL V. 39/49 (the largest complete individual, carapace approximately 49 cm long), and about 65% of *Proterochersis robusta* SMNS 17561 (the best preserved specimen, carapace about 34 cm long), which amounts to approximately 20–25 cm of total carapace length (the proportion of the bridge area to the carapace length seems to decrease with age in *Proterochersis porebensis*, although taphonomic deformation may also be at play). Despite that, and accounting for the preservation, the specimen lacks obvious juvenile characteristics (e.g., raised sulci borders [18]) and shows a relatively good sulci definition and scute features (e.g., development of marginal serration) – much better than ZPAL V. 39/34 and comparable to SMNS 17561. Size estimation is much more difficult for SM2015-1-001 and SM2017-1-129 due to the variability of gular and extragular size and lack of other unequivocal landmarks, but they likely belonged to individuals larger then SM2017-1-125, possibly comparable in size with SMNS 17561. The estimations calculated here for *Thaichelys ruchae* are somewhat smaller than those provided by Broin [2] (28–30 cm for SM2017-1-125 and up to 40 cm for SM2015-1-001), probably in part due to differing identification of shell landmarks and in part due to the much better sample of *Proterochersis* spp. specimens gathered and studied in recent years.

## Supporting information

**S1 Text. Extended information on the phylogenetic analysis – sources for scoring, changed scores, characters list, matrix, trees and lists of synapomorphies.**
(DOCX)

**S1 File. Phylogenetic matrix in Nexus format.**
(NEX)

**Fig S1. Strict consensus tree.** Above the branches node numbers, below bootstrap values above 50.
(PNG)

**Fig S2. Majority rule (50%) consensus tree.** Numbers above the branches indicate node numbers.
(PNG)

**S1 Archive.** ZIP archive with Figs 1–4 in full resolution.
(ZIP)

**S2 Archive.** ZIP archive with Figs 5–9 in full resolution.
(ZIP)

**S3 Archive.** ZIP archive with Figs 10–18 in full resolution.
(ZIP)

## Acknowlegments

We thank Gabriela Cisterna (PULR), Laura Cotton (NHMD), Sandra Daillie (MNHN), Zheng Fang (IVPP), Eudald Mujal Grané (SMNS), Nour-Eddine Jalil (MNHN), Bouziane Khalloufi (PRC), Chun Li (IVPP), Ning Li (IVPP), Sasa-On Khansubha (SM), Tida Liard (SM), Bent Erik Kramer Lindow (NHMD), Apirut Nilpanapan (PRC), Thanit Nonsrirach (PRC), Andrea Oettl (SMF), Khongsit Sawangsri (SM), Rainer Schoch (SMNS), Daniela Schwarz (MB), Rolf Schweizer (CSMM), Ron Son (SM), Heike Straebelow (MB), and Mongkol Udchachon (PRC), for the access to the specimens housed at their respective institutions, Torsten Scheyer (Paläontologisches Institut und Museum, Universität Zürich) for access to the remainder of the *Proganochelys quenstedtii* material from SMF studied by him, and Nils Natorp (Geocenter Møns Klint) and Volker Neipp (Museum Auberlehaus in Trossingen) for access to 'cf. *Proganochelys*' NHMD 190349 and *Proganochelys quenstedtii* SMNS 17204 on their expositions. We also thank Eric Buffetaut (French National Center for Scientific Research) for providing important information about the specimens and France de Lapparent de Broin for additional information and for notifying us about the new specimens in the MNHN collection. Milan Chroust is thanked for his help in photogrammetry of proterochersid shells housed in ZPAL. Tomasz Sulej (ZPAL) and all participants of the excavations in Poręba are thanked for their contribution to recovery of *Proterochersis porebensis* material. Finally, we thank the Reviewers, the Editor, and the Production Office for their time and work needed to process this manuscript.

## Author contributions

**Conceptualization:** Tomasz Szczygielski.

**Data curation:** Tomasz Szczygielski.

**Formal analysis:** Tomasz Szczygielski.

**Funding acquisition:** Tomasz Szczygielski.

**Investigation:** Tomasz Szczygielski, Dawid Dróżdż.

**Methodology:** Tomasz Szczygielski.

**Project administration:** Tomasz Szczygielski.

**Supervision:** Tomasz Szczygielski.

**Validation:** Tomasz Szczygielski, Phornphen Chanthasit, Sita Manitkoon, Pitaksit Ditbanjong.

**Visualization:** Tomasz Szczygielski, Dawid Dróżdż, Sita Manitkoon.

**Writing – original draft:** Tomasz Szczygielski.

**Writing – review & editing:** Tomasz Szczygielski, Dawid Dróżdż, Phornphen Chanthasit, Sita Manitkoon, Pitaksit Ditbanjong.

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
