## [Decision Letter · Decision Letter 0]

12 Jun 2024

PONE-D-24-13142The Triassic turtle of Thailand – revision of ‘Proganochelys’ ruchaePLOS ONE

Dear Dr. Szczygielski,

Thank you for submitting your manuscript to PLOS ONE. After careful consideration, we feel that it has merit but does not fully meet PLOS ONE’s publication criteria as it currently stands. Therefore, we invite you to submit a revised version of the manuscript that addresses the points raised during the review process.

We look forward to receiving your revised manuscript.

Kind regards,

Dawid Surmik, PhD

Academic Editor

PLOS ONE

Journal Requirements:

Reviewers' comments:

Reviewer's Responses to Questions

**Comments to the Author**

1. Is the manuscript technically sound, and do the data support the conclusions?

Reviewer #1: Yes

Reviewer #2: Yes

2. Has the statistical analysis been performed appropriately and rigorously? 

Reviewer #1: N/A

Reviewer #2: Yes

3. Have the authors made all data underlying the findings in their manuscript fully available?

Reviewer #1: No

Reviewer #2: Yes

4. Is the manuscript presented in an intelligible fashion and written in standard English?

Reviewer #1: Yes

Reviewer #2: Yes

5. Review Comments to the Author

Reviewer #1: Dear Editors, Dear Authors,

this manuscript provides the description of important fossil turtle material from the Late Triassic of Thailand and discussion of its evolutionary implications.

I find this to be a fantastic piece of work: the English is excellent, the descriptions detailed, the figures are clear, and all conclusions are thoughtful and based on the available data.

I marked up the attached manuscript with a small number of comments that the authors might wish to consider. The most important suggestions are as follows:

1) I would like to strongly urge the authors to upload all 3D models they obtained to a public repository such as MorphoSource. This will significantly raise the utility and impact of this work, as all people using their models will cite them as their source. I feel quite strongly about this detail, given that all data and models created in my own lab are made public.

2) Many electronic publications have the unfortunately tendency to heavily compress figures, whereby many details are lost. This can be compensated if the journal provides a link to images in full resolution or by uploading high resolution versions of all figures as supplements. Given the high quality of the figures being presented herein, I urge the authors to explore their possibilities with the present journal.

3) I think it is a good idea to phylogenetically define Proterochersidae, but suggest using Australochelys africanus as an additional external specifier. And while the authors are at it, they might as well define all three Triassic clade names: Proganochelyidae, Australochelyidae, and Proterochersidae. I provide additional comments in the manuscript.

I consider my comments to be minor and therefore see no need for another round of review. I strongly recommend this manuscript for publication.

The authors are welcome to know my identity.

Best regards,

Walter Joyce

Reviewer #2: This report seems very important for understanding the diversity of early turtles during the Triassic age, as materials from Asia must be so valuable in paleobiogeographical view points. I hope this will be published soon.

6. PLOS authors have the option to publish the peer review history of their article (what does this mean? ). If published, this will include your full peer review and any attached files.

**Do you want your identity to be public for this peer review?** For information about this choice, including consent withdrawal, please see our Privacy Policy .

Reviewer #1: No

Reviewer #2: **Yes: ** Ren Hirayama

---

## [Author Response · Author response to Decision Letter 1]

4 Dec 2024

Dear Editor, dear Reviewers,

Thank you for your comments and suggestions. We incorporated nearly all of them into the manuscript and corrected some typos and other small errors in the text and figure captions (marked with track changes). Below, we present responses (in bold) to individual points. Please excuse us for the delayed resubmission, but during the revision, we encountered in the collection of the Natural History Museum in Paris (MNHN) new, unpublished specimens of Thaichelys ruchae as well as casts of lost specimens. Among those is an isolated humerus – the only one currently known proterochersid appendicular bone aside from Proterochersis porebensis specimens. Because these finds are significant and supplement the available information on the anatomy and ontogenetic variability within the species, we decided to include them in the manuscript. Unfortunately, the MNHN collection was unavailable for some time due to the Olympics organized in Paris and then assignment of new specimen numbers by the collection manager took some time, which caused delays, but we hope that new data make the wait worth it.

Yours sincerely,

Tomasz Szczygielski

Reviewer #1

Dear Editors, Dear Authors,

this manuscript provides the description of important fossil turtle material from the Late Triassic of Thailand and discussion of its evolutionary implications.

I find this to be a fantastic piece of work: the English is excellent, the descriptions detailed, the figures are clear, and all conclusions are thoughtful and based on the available data.

Thank you very much, we appreciate that!

I marked up the attached manuscript with a small number of comments that the authors might wish to consider. The most important suggestions are as follows:

1) I would like to strongly urge the authors to upload all 3D models they obtained to a public repository such as MorphoSource. This will significantly raise the utility and impact of this work, as all people using their models will cite them as their source. I feel quite strongly about this detail, given that all data and models created in my own lab are made public.

Yes, this was not stated before but the models are uploaded to MorphoSource and the link to the MorphoSource project was added to the manuscript.

2) Many electronic publications have the unfortunately tendency to heavily compress figures, whereby many details are lost. This can be compensated if the journal provides a link to images in full resolution or by uploading high resolution versions of all figures as supplements. Given the high quality of the figures being presented herein, I urge the authors to explore their possibilities with the present journal.

The quality of figures in PLoS One is typically good but we included the figures in full resolution (600 DPI) in the supplement.

3) I think it is a good idea to phylogenetically define Proterochersidae, but suggest using Australochelys africanus as an additional external specifier. And while the authors are at it, they might as well define all three Triassic clade names: Proganochelyidae, Australochelyidae, and Proterochersidae. I provide additional comments in the manuscript.

We added Australochelys africanus as one of the specifiers. The clades Proganochelyidae and Australochelyidae will be defined in the upcoming manuscript describing a new proganochelyid from Greenland, which provides new relevant information.

I consider my comments to be minor and therefore see no need for another round of review. I strongly recommend this manuscript for publication.

Thank you very much!

The authors are welcome to know my identity.

Best regards,

Walter Joyce

This is a useful addition to the currently available framework of phylogenetic nomenclature. Thanks!

Page 18, lines 405–408: It seems that the best way to have turtles named after you is to describe material superficially and then block access to colleagues for decades (Francemys, Lapparentemys, etc.). Name as you wish, but is there really nobody more deserving? A place or person from Thailand? Khonkaenchelys? Thaiemys? Teakea (Thai for "old turtle")?

This particular work was helped by France de Lapparent de Broin, who aided us in access to the cast of lost specimens (TF 1440-5), informed us about the newly prepared material in the MNHN collection, authored the original description of the species, and was one of the most significant paleocheloniologists of the late 20th and early 21st century. Of course, we cannot speak for others, regarding their experiences. However, following the suggestion of the Reviewer, we conducted a poll among the coauthors and Thaichelys was elected as the more popular option.

Page 50, lines 1226–1229: I personally take issue with calling for dispersal between continents prior to the origination of continents. After all, there was no such thing as "Asian" "Europe" or "North America" prior to the breakup of Pangea. Indeed, as various purported stem-turtles are found across the globe, it seems to me that the stem lineage is a global phenomonon, not an Asian one that demands "dispersing" over non existant barriers...

On one hand, this is a valid argument, so we rephrased that part to refer more directly to the Triassic geography. On the other hand, what we meant here is the geographic distance and the fact that the clade had to originate somewhere, at a single point in space and time, even if it attained a global distribution soon after. This seems to be significant in the context of understanding of turtle origins and may provide clues as to where the immediate ancestors or relatives of turtles should be searched for. Even if these points of occurrence were not completely separated by an ocean, still some significant barriers (rivers, lakes, deserts, mountain ranges, climate, habitat availability) could exist and limit the dispersal of the earliest turtles – note that the part of modern-day Asia including China and Thailand was in fact partially separated in the Triassic from the rest of Pangaea by the (Paleo-)Tethys, the land bridge between these parts was located at high latitudes and mountainy, possibly resulting in harsh environmental condition, and terrestrial (and even aquatic continental) vertebrates are usually quite susceptible to those factors. The spotty pattern of occurrence of Triassic turtles (present in abundance in some localities but completely absent in the others, even within the same formation) may indicate that they were pretty picky when it comes to the environmental setting.

Reviewer #2:

This report seems very important for understanding the diversity of early turtles during the Triassic age, as materials from Asia must be so valuable in paleobiogeographical view points. I hope this will be published soon.

Thank you!

---

## [Editor Report · Decision Letter 1]

10 Dec 2024

The Triassic turtle of Thailand – revision of ‘Proganochelys’ ruchae

PONE-D-24-13142R1

Dear Dr. Szczygielski,

We’re pleased to inform you that your manuscript has been judged scientifically suitable for publication and will be formally accepted for publication once it meets all outstanding technical requirements.

Kind regards,

Dawid Surmik, PhD

Academic Editor

PLOS ONE
---

## [Editor Report · Acceptance letter]

PONE-D-24-13142R1

PLOS ONE

Dear Dr. Szczygielski,

I'm pleased to inform you that your manuscript has been deemed suitable for publication in PLOS ONE. Congratulations! Your manuscript is now being handed over to our production team.

Kind regards,

on behalf of

Dr. Dawid Surmik

Academic Editor

PLOS ONE